# Effects and Mechanisms of China's Pilot Free Trade Zones on Green and High-Quality Development from the Dual-Circulation Perspective

Yanjun Chang * and Liuliu Lai

School of Economics and Management, Shanghai Institute of Technology, Shanghai 201418, China
* Correspondence: changyanjun@sit.edu.cn; Tel.: +86-136-0198-2823

**Abstract:** Under the dual pressure of economic growth and environmental pressure, green and high-quality development become a critical strategy for China to achieve win-win development for both the economy and the environment. In the dual-circulation pattern, a deep and high level of openness is the primary prerequisite for encouraging green and high-quality development. Previous studies have overlooked the crucial pivotal role of pilot free trade zones (PFTZs) in domestic and international circulation. The PFTZ is essential in driving China's green modernization and high-quality development as a crucial area of opening up. This paper utilizes the propensity score matching and difference-in-differences (PSM-DID) on panel data from 27 provinces from 2007 to 2020 to investigate the green policy effects of PFTZ building and its underlying mechanism. The results show that (1) the establishment of PFTZs significantly enhances regional green and high-quality development, with a net policy impact of roughly 4%. The several robustness tests demonstrate that the outcomes are extremely reliable; (2) According to the heterogeneity analysis, compared with inland areas, coastal locations may more effectively foster green and high-quality development; and (3) The influencing mechanism test results show that PFTZs primarily have a positive impact on technological progress, scale expansion, and innovation-driven effects, which, in turn, support green and high-quality development. The mediating effect of innovation-driven and technological progress is extremely significant among them. These research findings can provide empirical support and policy rationale for PFTZs to better promote China's green and high-quality development.

**Keywords:** pilot free trade zones; green and high-quality development; PSM-DID; innovation-driven; quasi-natural experiment

## 1. Introduction

China has achieved a miracle of rapid economic growth in the 40 years since it began to reform and open up. The quality of China's economic development has been constrained, however, by conventional economic development strategies including imbalanced, inefficient production and environmental degradation. The imbalance between the distribution of production factors and green development is the root of the issue. According to BP World Energy Statistics, China is the world's greatest energy user and carbon emitter, contributing 30.7% of the total carbon emissions in worldwide carbon emissions, of which primary energy consumption accounted for 26.5% [1]. Under the dual pressure of economic growth and environmental pressure, optimizing the economic structure with green development has become a crucial countermeasure for China to address this issue [2]. In the next 40 years, China will move into a new phase of development that is focused on sustainability, energetically pursuing the path of high-quality and environmentally friendly development.

The central strategic notion of China's development in the new era is to "promote high-quality development while accelerating the creation of the new development pattern." Adhering to a high level of openness is the primary prerequisite for the strategy's

implementation. As the frontier of China's reform and opening up, the construction and development of pilot free trade zones (PFTZs) offers strong power support for China's extensive and high level of opening up to the outside world. Six rounds of pilot building have been finished and 21 PFTZs have been formed in stages since the Shanghai PFTZ was founded in 2013 (Figure 1). On fewer than four thousandths of the country's surface area, by the end of 2021, PFTZs had brought in 18.5% of foreign investment and 17.3% of imports and exports. The Ministry of Ecology and Environment, together with various departments, issued the "Guidelines on Strengthening Ecological Environmental Protection in Pilot Free Trade Zones to Promote High-Quality Development" in May 2021. The opinion emphasizes the need for PFTZs to continue deepening innovation and focus on ecological preservation in order to play a crucial role in supporting high-quality and environmentally friendly development. Therefore, the significant practical issue that has to be resolved in the new growth pattern is how to play the key node of PFTZs effectively to open up the domestic and international dual circulation and better promote green and high-quality development.

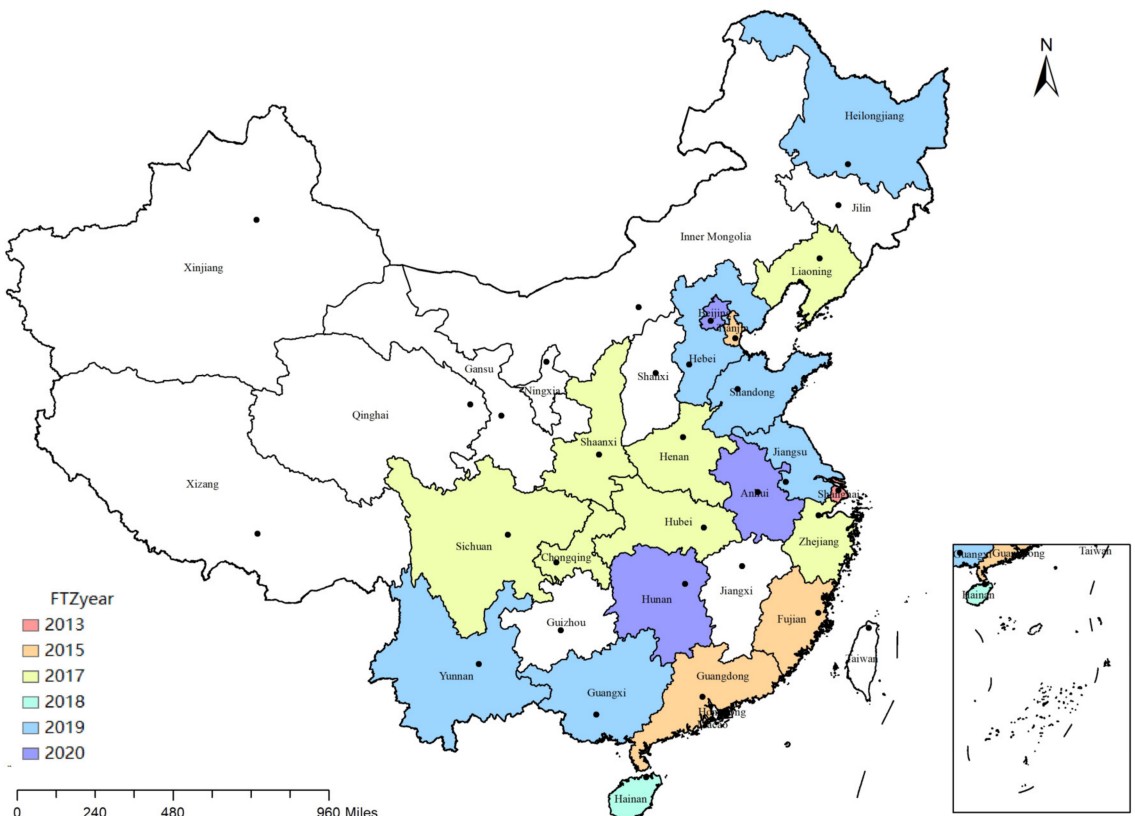

**Figure 1.** China's PFTZ construction. Note: The authors map the distribution of each PFTZ according to the general scheme of PFTZ construction issued by the State Council. The map data are obtained from the National Platform for Common Geospatial Information Services.

With institutional innovation at its center, PFTZ is a policy test zone that offers a high level of openness, including trade facilitation, investment liberalization, and financial internationalization [3]. There has been a fair amount of research on the policy evaluation of the economic consequences of PFTZs, with the main topics being economic growth [4–6], foreign direct investment [7–9], import and export trade [10,11], and financial innovation [12,13]. The findings of different scholars are largely consistent in that the economic effects of PFTZs are significantly positive. Research on the environmental impacts of PFTZ policies or the evaluation of green development, however, remains equivocal. Positive and negative points of view can be widely distinguished among the major ones. The PFTZs' development, in the opinion of the academics that Zhuo et al. are representing, is a "policy trap." That is, despite promoting rapid economic expansion, PFTZs can cause

severe environmental externalities. Using the Guangdong PFTZ in China as an example, Zhuo et al. [14] came to the empirical conclusion that for every 100 million yuan rise in GDP, there is an increase in 1.746 million tons of wastewater and 20.016 tons of waste gas. According to Tachie et al. [15], commercial openness may be a factor in the forced contamination of the environment. Although trade openness fosters economic expansion, it also worsens energy use and raises $CO_2$ emissions. He [16] also confirmed that the impact of FDI on economic growth causes an increase in industrial $SO_2$ emissions. Wang et al. [17] verified that the construction of PFTZs exacerbated the environmental pollution problem in the test area by using urban panel data.

On the other hand, some academics have advanced the other viewpoint, contending that the development of PFTZs has beneficial consequences on green policy. By analyzing the Shanghai PFTZ, Jiang et al. [18] discovered that PFTZ construction can considerably contribute to the growth of green total factor productivity in Shanghai. Ma et al. [19] and Wang et al. [20] suggested that PFTZ policies have a positive contribution to urban green production, where technological progress is the main driver. Erdogan [21] created a trade environment model and came to the conclusion that trade liberalization makes it easier to cut back on local or national pollution emissions. Li et al. [22] used the Belt and Road Initiative to measure how open policies affected the environment. The results of the study demonstrated that the open policies significantly increased the level of regional green development and helped build a sustainable ecological environment in the provinces and nations along the route.

According to the existing research, scholars affirm that PFTZ building has a considerable policy impact on promoting economic growth, although there are still some questions concerning its environmental effects. Some academics have overlooked the dialectical connection between economic quality and environmentally friendly growth, as well as the crucial significance of PFTZ construction in the dual domestic and international circulation. The growth of PFTZs in the past was prioritized above the caliber of development in order to seek the development pace of foreign direct investment and commerce access to the outside world. Environmental benefits are considerably diminished by development models that need large inputs of energy [23,24]. PFTZs emphasize green orientation and the quality and efficiency of factor flow in the new growth pattern, and its crucial essential role is becoming more noticeable. As the link between domestic and foreign commerce, PFTZs serve as a catalyst for internal and external circulation and a crucial groundwork for the advancement of high-quality green trade [2]. Will the creation of PFTZs therefore considerably enhance green and high-quality development? Does the impact of PFTZ construction on green policies vary by region? What is the influencing mechanism by which PFTZ building affect green and high-quality development? The solutions to these queries have significant theoretical and practical ramifications for investigating PFTZs to support green and high-quality development.

In light of this, this paper considers PFTZs as a quasi-natural experiment that utilizes the propensity score matching and difference-in-differences method (PSM-DID) to evaluate the policy effects of PFTZ construction on green and high-quality development. The main research contributions of this paper include the following: (1) Considering the key linkage role of PFTZs in the new development pattern of dual circulation, this paper constructs a comprehensive evaluation system of green and high-quality development in four aspects, including internal and external circulation, economic development, and environment, so as to quantify the quality of green development in each region. Additionally, in order to investigate the causal connection between PFTZs and green high-quality development, this paper incorporates both into the same research framework; (2) Current studies mostly evaluate the design of PFTZs using the twofold difference approach, but they are unable to address the issue of selective bias in the estimated sample. This paper uses PSM-DID for regression analysis to effectively eliminate the influence of unobservable factors and thus obtain the true net effect of policy implementation; and (3) The influence of PFTZs on environmental consequences is investigated individually at the level of the mechanism in

comparison to earlier work. This paper explains the internal driving paths of the policy effects and theoretically and empirically investigates the various implications of PFTZs on sustainable and high-quality development.

The remainder of the paper is structured as follows: Section 2 presents the research hypotheses based on theoretical analysis. Section 3 is the research design, which introduces the model construction, indicator selection, and data sources. Section 4 is the empirical analysis, which mainly discusses, from benchmark regression, a series of robustness tests and heterogeneity analysis. Section 5 examines the influencing mechanism of PFTZs on green and high-quality development. Finally, Section 6 summarizes the findings of this study and proposes corresponding policy recommendations.

## 2. Theoretical Analysis and Hypothesis

The Environment Kuznets Curve (EKC) hypothesis is the classic theoretical model for measuring the relationship between economic development and environmental quality. Researchers summarize the intrinsic influence mechanism of the EKC hypothesis based on its viability [25–27]. Specifically, the scale, structure, and technology effects of a region's economic development have an impact on that region's environmental quality. While, under free trade, the dynamic link between economic activity and environmental quality is even more pronounced [28]. The main goal of establishing PFTZs is to provide China with the expertise necessary to create a new open economic system through system innovation. Therefore, promoting trade liberalization is the key task for the construction and development of PFTZs. Based on this, the EKC hypothesis put forward by Grossman and Krueger [29] can be used to explain the theoretical mechanism of PFTZ building on a regional green and high-quality development. According to the functional positioning of PFTZs in the dual-circulation development pattern, this paper integrates EKC theory and related scholars' research to analyze the internal mechanism of policy effects from three paths: technological progress, scale expansion, and innovation-driven effects.

The technological progress effect refers to the construction of PFTZs by influencing the level of technology and generating technological change, which, in turn, affects the quality of green development. By increasing investment in technical R&D and securing original technological achievements, enterprises in the PFTZs can advance technology thanks to the clear, green development-oriented requirements. Fei et al. [30] pointed out that, in an open economy, independent innovation can continuously raise the standard of regional green development. Likewise, Anser et al. [31] demonstrate how the results of technological progress contribute to the achievement of sustainable development goals. Therefore, the establishment of PFTZs can improve the quality of local green economy development through the technological progress effect.

The scale expansion effect describes how the building of PFTZs affects the scale of economic activity in the area and, consequently, green high-quality development. Relying on institutional advantages, PFTZs provide a liberalized and facilitated policy environment for trade, investment, and financial innovation. This can directly alleviate difficulties such as lack of capital in the process of local economic development and promote the expansion of the economy. Economic scale is the significant factor affecting green high-quality development [32,33]. Trade liberalization has a bigger impact on environmental quality in regions with larger economies. Thus, through the scale expansion effect, the creation of PFTZs can enhance the quality of local green economic growth.

The innovation-driven effect refers to the impact of PFTZ construction on green high-quality developments by influencing the regional innovation environment. According to Lee and Wang [33], lowering environmental pollution requires a conducive climate for innovation. With more palatable trade and investment rules, PFTZs draw in high-quality foreign money and expertise, enabling local businesses to innovate and expand while also indirectly enhancing the ecosystem for innovation. Lin and Ma [34] argued that the environment for innovation supports the flow of elements necessary for the development of green technologies, assisting in the balancing of economic quality and environmentally

friendly development. As a result, PFTZs' innovation-driven effect can enhance regional green high-quality development.

Based on the above analysis, this paper proposes the following research hypothesis:

**Hypothesis 1 (H1).** *The establishment of PFTZs can significantly promote regional green and high-quality development.*

**Hypothesis 2a (H2a).** *PFTZs can enhance green and high-quality development through the technological progress effect.*

**Hypothesis 2b (H2b).** *PFTZs can enhance green and high-quality development through the scale expansion effect.*

**Hypothesis 2c (H2c).** *PFTZs can enhance green and high-quality development through the innovation-driven effect.*

### 3. Research Design

*3.1. Model Specification*

The difference-in-differences method (DID) is the mainstream research method currently used to assess the effect of a certain policy. Direct application of DID may result in biased estimation findings because of the wide variation between provinces. Propensity score matching (PSM) can address sample selection bias, but not endogeneity due to the omission of variables. Based on this, this paper draws on the study of [35,36] to combine the two to more scientifically and accurately evaluate policy effects.

Based on PSM, the control group that resembles the experimental group the most is identified. In the first five batches of PFTZ building, the experimental group of provinces is discussed in this study. The policy effects of the sixth set of PFTZs, which includes Beijing, Hunan, and Anhui, are not taken into account as yet because they were founded late. We use logit regression to obtain propensity score values for sample matching. Then, the samples are examined for balance to see if there was a significant difference between the matched treatment and control groups. The randomness of sample selection is ensured to eliminate selection bias.

$$P(X_i) = Pr[T_i = 1|X_i] = \frac{exp(bX_i)}{1 + exp(bX_i)} \tag{1}$$

in model (1), the treatment group represents the provinces that build PFTZs, denoted as $T_i = 1$, while the control group represents non-PFTZs, denoted as $T_i = 0$. $X$ denotes the matching variables, which represent the factors that influence the selection of sample $i$ into the experimental group.

Based on PSM results, the net effect generated by the policy is tested using *DID*. Since there are temporal batch differences in the construction of PFTZs nationwide, this paper refers to Beck et al. [37] to construct the benchmark model using multi-period DID.

$$GHD_{it} = \alpha_0 + \alpha_1 DID_{it} + \gamma X_{it} + \mu_i + f_t + e_{it} \tag{2}$$

$$DID_{it} = Treat_i \times Post_{it} \tag{3}$$

$i$ denotes province, $t$ denotes year, and $GHD_{it}$ is the explained variable in this paper, representing the level of green and high-quality development. $Treat_i$ denotes the dummy variable of whether the policy is implemented or not. In this study period, $Treat_i = 1$ if PFTZ is built in region $i$; otherwise, $Treat_i = 0$. $Post_{it}$ denotes the time of PFTZ policy implementation. If region i builds PFTZ in t-th year, the policy takes the value of 1 for the year of implementation and subsequent years, and 0 for the rest. $DID_{it}$ is the policy dummy variable formed by the interaction term between the two, i.e., the core explanatory

variable. $\alpha_1$ denotes the policy treatment effect of PFTZs on regional green high-quality development, which is an important parameter of interest in this paper. If it is significantly positive, it proves that the construction of PFTZs can promote the green and high-quality development of the province where it is located. $X_{it}$ represents all control variables. $\mu_i$ stands for individual-fixed effects. $f_t$ stands for time-fixed effects. $e_{it}$ is the standard error.

### 3.2. Variable Definition and Data Sources

Green high-quality development is the explained variable. With the faster implementation of the new growth pattern, China's economic and social development has given more attention to the quality and efficiency of green factor flows than most previous research on green high-quality development achieved [36,38–40]. Therefore, this paper draws on the study of Wang et al. [41] to construct a green and high-quality development index (GHD) to represent the quality of regional green development by using the entropy weighting method. Table 1 shows the description and weights of relevant indicators.

**Table 1.** Green and high-quality development index.

| Dimension Layer | Sub-Level | Explanation | Attributes | Weights |
|---|---|---|---|---|
| Economic development | GDP growth rate | GDP growth rate | Positive | 0.010 |
| | Education level | Ratio of education spending | Positive | 0.034 |
| | Communication infrastructure | Fixed asset investment in communication services | Positive | 0.085 |
| Internal circulation | Production | Ratio of new product sales revenue to main business revenue | Positive | 0.083 |
| | Distribution | Ratio of disposable income per capita to GDP | Positive | 0.044 |
| | | Ratio of total retail sales of consumer goods to GDP | Positive | 0.031 |
| | Exchange | Ratio of added value of transportation, storage, and postal industry to GDP | Positive | 0.040 |
| | Consumption | Ratio of per capita consumption expenditure to per capita GDP | Positive | 0.052 |
| | | Final consumption rate | Positive | 0.049 |
| External circulation | Foreign trade dependence | Ratio of total import and export trade to GDP | Reverse | 0.011 |
| | Two-way investment | Ratio of foreign direct investment to GDP | Positive | 0.203 |
| | | Ratio of outward foreign direct investment to GDP | Positive | 0.239 |
| Environment | Environmental regulation | Ratio of completed investment in industrial pollution control to the value-added of the secondary industry | Positive | 0.109 |
| | Unit output of wastewater | Ratio of total wastewater discharge to GDP | Reverse | 0.005 |
| | Unit output of exhaust gas | Ratio of sulfur dioxide emissions to GDP | Reverse | 0.004 |

Based on the availability of data, the following control variables are selected in this paper with reference to previous literature: value-added of secondary industry and value-added of tertiary industry are used to characterize the development of local industries. Government support is measured by the ratio of regional fiscal expenditure. The level of financial deepening is measured by the ratio of total deposits and loans of financial institutions to GDP. The ratio of tax on the main business to a profit of industrial enterprises above the scale indicates the tax burden of enterprises. Energy consumption is expressed as the ratio of coal consumption. Some of the variables with large values are taken as natural logarithms to ensure the smoothness of the sample data.

The data used in this paper are the panel data of 27 provinces in China from 2007 to 2020. The relevant raw data are mainly derived from public information such as the China Statistical Yearbook, the provincial statistical yearbooks, and the statistical bulletins on economic and social development in previous years. Table 2 demonstrates the specific description and descriptive statistics of the variables.

**Table 2.** List of explanations of variables.

| Variable | Abbreviation | Explanation | Mean | Std. Dev. | Min | Max |
|---|---|---|---|---|---|---|
| Green and high-quality development | GHD | Comprehensive evaluation index system | 0.180 | 0.043 | 0.110 | 0.410 |
| Pilot free trade zone | PFTZ | Dummy Variable (0 or 1) | 0.183 | 0.387 | 0.000 | 1.000 |
| Value-added of secondary industry | lnsecond | Value-added of secondary industry, taking the logarithm | 8.764 | 1.003 | 5.898 | 10.698 |
| Value-added of tertiary industry | lnthrid | Value-added of secondary industry, taking the logarithm | 8.733 | 1.033 | 5.694 | 11.044 |
| Government support | GS | Regional budget expenditures (% of GDP) | 0.255 | 0.115 | 0.100 | 0.760 |
| Financial deepening | FD | The total deposits and loans of financial institutions (% of GDP) | 2.942 | 0.853 | 1.400 | 6.260 |
| Enterprise tax revenue | ETR | Taxes and extra charges on the main business of industrial enterprises above the designated size (% of total profit) | 0.965 | 0.422 | 0.340 | 2.500 |
| Energy consumption | EC | coal consumption (% of total energy consumption) | 0.965 | 0.422 | 0.340 | 2.500 |

## 4. Empirical Results

### 4.1. Propensity Score Matching (PSM)

According to the research design, radius matching is used in this paper. The difference in score values between the experimental and control groups was set within r = 0.05 for matching. Tables 3 and 4 present the results of the balance test for PSM. The results indicate that in both the experimental and control groups, the matched variables' absolute standard deviation is less than 20%. After matching, none of the variables' findings from the t-test or the LR test were significant. Figure 2 indicates the standard deviation before and after matching the covariates. This shows that there is no significant difference between the matched experimental and control groups.

**Table 3.** Balance test of variables before and after PSM.

| Variables | Sample | Mean | | % Bias | % Reduct | *t*-Test | |
|---|---|---|---|---|---|---|---|
| | | PFTZ | Non-PFTZ | | \|Bias\| | t | *p* > \|t\| |
| lnsecond | U | 9.114 | 8.0632 | 122.9 | | 11.03 | 0.000 |
| | M | 8.5021 | 8.6528 | −17.6 | 85.7 | −1.35 | 0.179 |
| lnthrid | U | 9.1152 | 7.9589 | 130.2 | | 11.42 | 0.000 |
| | M | 8.4478 | 8.5113 | −7.2 | 94.5 | −0.62 | 0.539 |
| GS | U | 0.2048 | 0.3550 | −149.6 | | −15.19 | 0.000 |
| | M | 0.2610 | 0.2492 | 10.5 | 93.0 | 1.21 | 0.226 |
| FD | U | 2.9339 | 2.9594 | −3.0 | | −0.27 | 0.784 |
| | M | 2.7299 | 2.6201 | 13.1 | −329.4 | 1.12 | 0.263 |
| ETR | U | 0.3771 | 0.5472 | −22.5 | | −2.39 | 0.018 |
| | M | 0.4971 | 0.5201 | −3.0 | 86.5 | −0.19 | 0.850 |
| EC | U | 0.8133 | 1.2680 | −113.0 | | −11.44 | 0.000 |
| | M | 0.9200 | 0.9551 | −8.7 | 92.3 | −0.96 | 0.339 |

Note: U refers to the sample before employing the PSM, while M denotes the matched sample after employing the PSM.

**Table 4.** Balance test of samples before and after PSM.

| Sample | Pseudo-R2 | LR Test | *p*-Value | B (%) |
|---|---|---|---|---|
| U | 0.697 | 335.21 | 0.000 | 224.3 |
| M | 0.019 | 4.68 | 0.586 | 31.7 |

Note: U refers to the sample before employing the PSM, while M denotes the matched sample after employing the PSM. Pseudo-R2 implies the goodness-of-fit of logit regression. *p*-value provides the significance probability value of the LR test. B (%) denotes the standardized bias difference between the unmatched and matched samples.

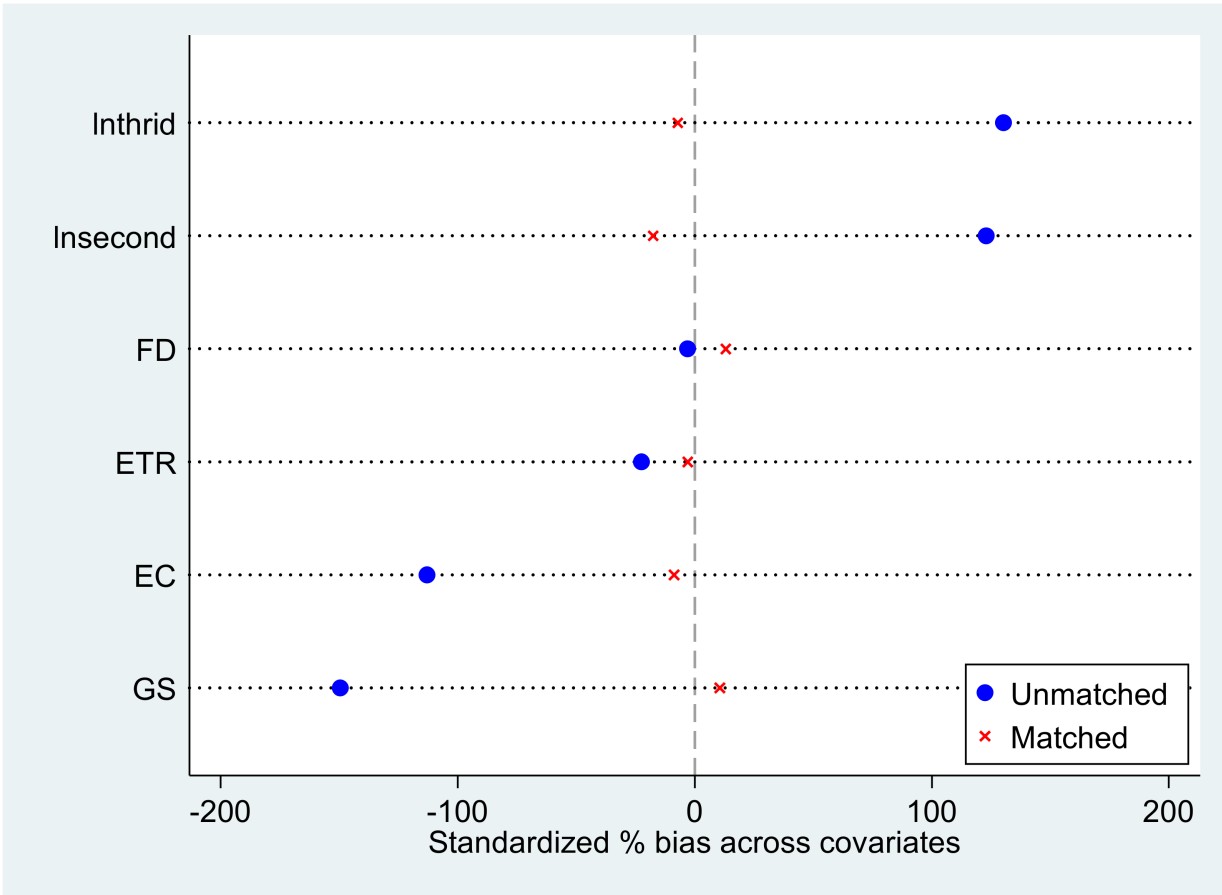

**Figure 2.** The standard deviation before and after matching the covariates.

### 4.2. Parallel Trend Assumption of DID

The prerequisite assumption for using DID is that the experimental group and the control group have non-varying differences over time before the policy is implemented. That is, both have the same trend before the policy implementation.

This paper winsorizes for the year of policy implementation in order to address the issue of tiny sample values at both ends due to the sample time span of 12 years. Simultaneously, 1 year before the policy implementation was selected as the benchmark group to assess the changing trend of green and high-quality development in the experimental and control groups. Figure 3 shows that there was no regularity in the regression coefficient changes prior to the policy shock, and none of the coefficients were significant at the 95% level of the confidence interval. Therefore, it can be concluded that before the implementation of the PFTZ policy, there were no appreciable differences between the changes in the experimental and control groups. This is in line with the parallel trend with multi-period DID.

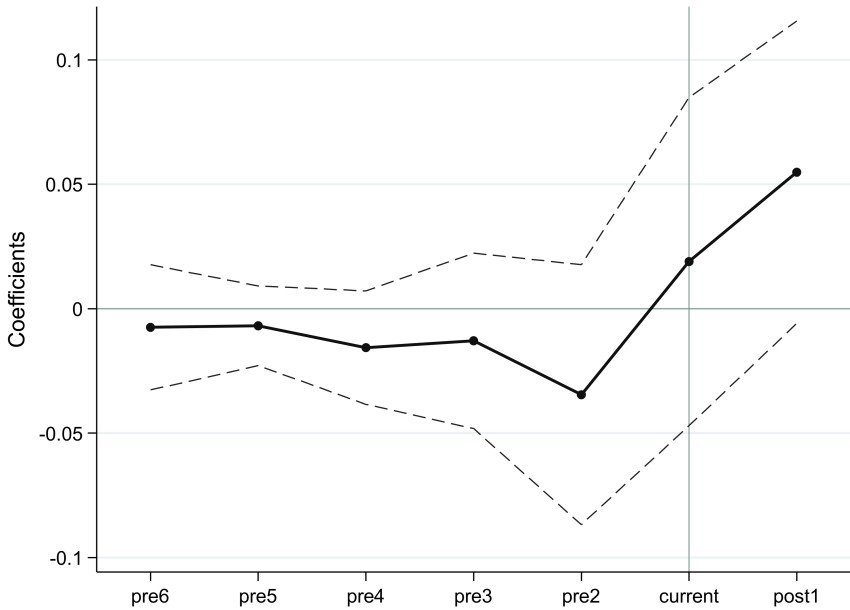

**Figure 3.** Parallel trend test.

*4.3. Benchmark Regression Analysis*

Table 5 presents the results of the test of PFTZ construction on green and high-quality development. Column (1) reports the regression coefficients without considering fixed effects. Column (2) adds control variables for regression on the basis of column (1). After accounting for year and province-fixed effects, column (3) uses samples matched according to sample weights. The samples in column (4) meet the common support assumption. The findings reveal that the core explanatory variable's coefficients are consistently significantly positive despite the various variables taken into account, demonstrating that PFTZ construction can promote local green and high-quality development. Specifically, if a province builds a PFTZ, then the quality of local green development will increase by 4%. H1 is verified.

**Table 5.** PSM-DID baseline regression results.

|  | (1) | (2) | (3) | (4) |
|---|---|---|---|---|
| DID | 0.0343 ** | 0.0285 * | 0.0213 * | 0.0397 ** |
|  | (0.0134) | (0.0117) | (0.0112) | (0.0175) |
| lnsecond |  | −0.0539 *** | −0.0718 *** | −0.0545 *** |
|  |  | (0.0155) | (0.0174) | (0.0118) |
| lnthrid |  | 0.0399 ** | 0.1091 *** | 0.0906 *** |
|  |  | (0.0177) | (0.0179) | (0.0272) |
| GS |  | −0.1503 | −0.0170 | 0.0712 |
|  |  | (0.1125) | (0.1569) | (0.2064) |
| FD |  | −0.0045 | 0.0054 | 0.0185 |
|  |  | (0.0072) | (0.0108) | (0.0123) |
| ETR |  | 0.0026 | −0.0048 | −0.0135 ** |
|  |  | (0.0061) | (0.0029) | (0.0052) |
| EC |  | −0.0606 *** | −0.0293 | −0.0476 * |
|  |  | (0.0125) | (0.0194) | (0.0248) |
| Constant | 0.1694 *** | 0.3993 *** | −0.1269 | −0.1487 |
|  | (0.0041) | (0.0450) | (0.0980) | (0.1793) |
| $R^2$ | 0.0529 | 0.5231 | 0.7889 | 0.7061 |
| N | 119 | 119 | 119 | 119 |
| Year-fixed effect | No | No | Yes | Yes |
| Province-fixed effect | No | No | Yes | Yes |

Note: *, **, and *** indicate that the statistical value is significant at 10%, 5%, and 1%, respectively. Values in parentheses are standard deviations.

### 4.4. Regional Heterogeneity Analysis

In terms of their potential for internal and external recycling as well as environmental protection, different provinces may have varying levels of economic growth. In light of this, this paper divides PFTZs into coastal and inland to test the geographical layout and regional heterogeneity. The results in Table 6 show that the impact of PFTZ construction on coastal areas is significantly positive, while the impact on inland PFTZs is not significant. This indicates that there is heterogeneity in the impact of geographical location on green and high-quality development. Specifically, compared with inland areas, PFTZ policy is more favorable to green and high-quality development in coastal areas. The main reason for this is the differences in industrial structure, government governance level, and business environment in coastal areas.

**Table 6.** Heterogeneity tests.

|  | **(1)** | **(2)** |
|---|---|---|
| DID_coastal | 0.0622 *** | |
|  | (0.0153) | |
| DID_inland | | 0.0056 |
|  | | (0.0283) |
| lnsecond | −0.0498 *** | −0.0388 ** |
|  | (0.0101) | (0.0158) |
| lnthrid | 0.0639 ** | 0.0585 ** |
|  | (0.0240) | (0.0262) |
| GS | −0.0394 | 0.2231 |
|  | (0.2097) | (0.3102) |
| FD | 0.0192 | 0.0020 |
|  | (0.0163) | (0.0142) |
| ETR | −0.0095 * | −0.0155 * |
|  | (0.0057) | (0.0085) |
| EC | −0.0192 | −0.0265 |
|  | (0.0137) | (0.0266) |
| Constant | 0.0356 | −0.0214 |
|  | (0.1961) | (0.1506) |
| $R^2$ | 0.7178 | 0.6817 |
| N | 118 | 118 |
| Year-fixed effect | Yes | Yes |
| Province-fixed effect | Yes | Yes |

Note: *, **, and *** indicate that the statistical value is significant at 10%, 5%, and 1%, respectively. Values in parentheses are standard deviations.

### 4.5. Robustness Tests

The exclusion of additional unobservable variables from interfering with the estimation results is another crucial aspect of the experimental validity of the multi-period DID. Despite the inclusion of control variables in the baseline regression, there may still be some unobserved variables that have an impact on the estimation results of this paper. We randomly altered the experimental group of individuals in terms of the timing of policy shocks in order to test the robustness of the regression results. The placebo test for 500 repeated regressions is shown in Figure 4, randomly generated estimated coefficients approximating a normal distribution with a mean of 0. This indicates that the experimental results of this paper were not influenced by other policy or randomness factors.

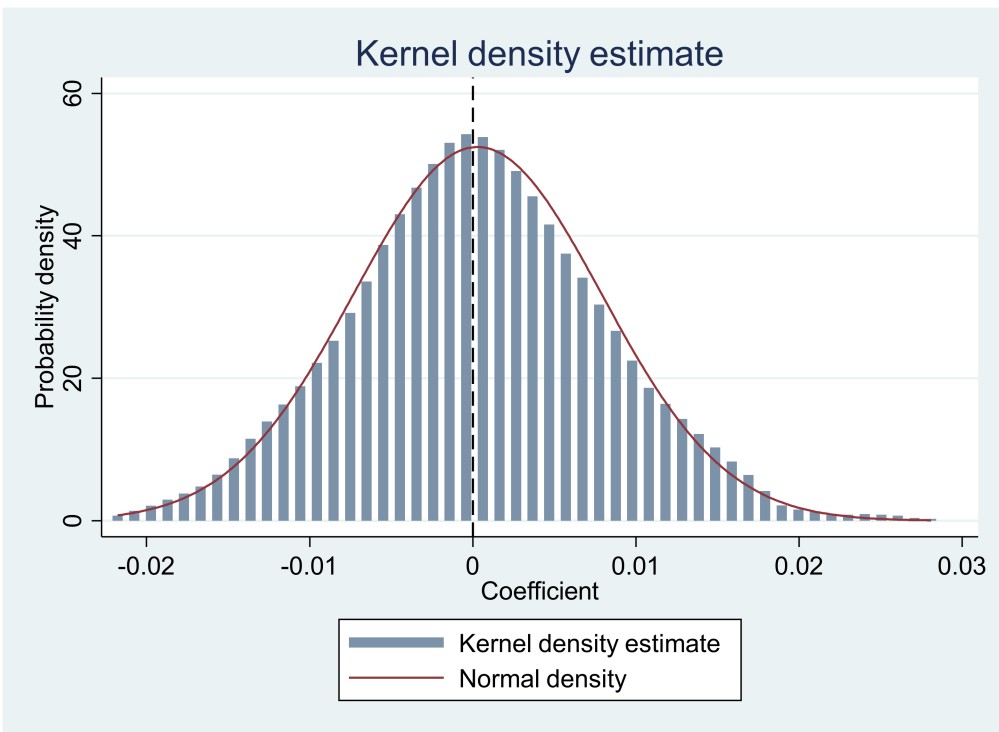

**Figure 4.** Placebo test.

To further test the robustness of the estimation results, this paper is re-regressed using the exclusion of SEZs and counterfactual analysis, respectively. Column (1) of Table 6 displays the regression results of model (2) without SEZs. Special Economic Zones (SEZs) are the outcome of innovation in opening up to the outside world at the beginning of reform, while PFTZs are a test bed for transforming traditional SEZs and achieving an open economy. Considering the endogeneity problem caused by SEZs, this paper excludes the regions where SEZs are established for testing. The findings demonstrate that even after SEZs are excluded, PFTZs still have a significant policy effect. H1 is supported.

Second, by erroneously changing the PFTZ's establishment time, this research performs robustness tests. To provide policy dummy variables for testing, the establishment time is artificially advanced by 1–3 years, correspondingly. If the estimated coefficients of the interaction terms are not significant, the inverse suggests that the finding that the establishment of PFTZs promotes green and high-quality development holds. Columns (2)–(4) of Table 7 show that none of the coefficients of the interaction terms based on the spurious setting of the timing of the establishment of PFTZs are significant. That is, the reverse supports the baseline regression results.

**Table 7.** Robustness tests.

|  | (1) | (2) | (3) | (4) |
|---|---|---|---|---|
| DID | 0.0362 ** |  |  |  |
|  | (0.0134) |  |  |  |
| DID_Before1 |  | 0.0428 |  |  |
|  |  | (0.0244) |  |  |
| DID_Before2 |  |  | 0.0056 |  |
|  |  |  | (0.0082) |  |
| DID_Before3 |  |  |  | 0.0109 |
|  |  |  |  | (0.0110) |
| Controls | Yes | Yes | Yes | Yes |
| Constant | −0.1917 | −0.1553 | −0.0431 | −0.0256 |
|  | (0.2040) | (0.1993) | (0.1409) | (0.1538) |

**Table 7.** *Cont.*

|          | (1)    | (2)    | (3)    | (4)    |
| :------: | :----: | :----: | :----: | :----: |
| $R^2$    | 0.7491 | 0.7168 | 0.6821 | 0.6837 |
| N        | 103    | 118    | 118    | 118    |
| Year     | Yes    | Yes    | Yes    | Yes    |
| Province | Yes    | Yes    | Yes    | Yes    |

Note: ** indicates that the statistical value is significant at 5%. Values in parentheses are standard deviations.

## 5. Influencing Mechanism Analysis

From the previous empirical findings, it is clear that PFTZ implementation significantly promotes green and high-quality development. So how do PFTZs support the regional green economic quality? As analyzed in the previous section on theoretical mechanisms, PFTZs have technological progress, scale expansion, and innovation-driven effects that can influence green and high-quality development through these three paths. Based on this, this paper draws on Baron and Kenny [42] to construct a mediating effects model on the basis of model (2) and apply the Sobel test and the Bootstrap test to verify the existence of the above mechanism. The specific model is set as follows.

$$Med_{it} = \beta_0 + \beta_1 DID_{it} + \delta X_{it} + \mu_i + f_t + \varepsilon_{it} \tag{4}$$

$$GHD_{it} = \gamma_0 + \gamma_1 DID_{it} + \gamma_2 Med_{it} + \delta X_{it} + \mu_i + f_t + \varepsilon_{it} \tag{5}$$

Model (4) reflects the effect of PFTZ construction on the mediating variables. $Med_{it}$ is a set of mediating variables. That is, the technological progress, scale expansion, and innovation-driven effects mentioned in H2a–H2c. These are denoted as technological progress (inno), green finance (gf), and industry–university–research cooperation (iur), respectively. Other variables are defined in the same way as in model (1).

Model (5) reflects the simultaneous effects of PFTZ construction and mediating variables on regional green and high-quality development. The estimated parameters $\gamma_1$ are the direct effects of PFTZ construction on green and high-quality development, and $\gamma_2$ is the effect of mediating variables. Other variables are defined in the same way as model (1).

### 5.1. Technological Progress Mediating Effects

This paper uses the number of granted invention patents per 10,000 people to assess the level of regional technological development, relying on the research of [43,44]. Columns (1) to (2) of Table 8 display the specific empirical findings. Column (1) shows that the coefficient of the relationship between PFTZ construction and technological progress is significantly positive. This indicates that the implementation of PFTZ policy effectively promotes the breakthrough innovation of technological activities in the region. The results in column (2) show that the regression coefficient of PFTZs is significantly positive at the 10% level when technical progress is included in the model, while the regression coefficient of technical progress is not significant. This research employs the Sobel test and the Bootstrap test, respectively, to confirm if the mediating impact of technical progress affecting green and high-quality development occurs. The results in Table 9 show that the *p*-value of the Sobel test is 0.0044, which is less than 0.05. The confidence interval of the Bootstrap test is [0.0183,0.0434], which does not contain 0. This implies that there is a mediating influence of technological progress and supports H2a.

**Table 8.** Influencing mechanism estimation results.

| | (1) | (2) | (3) | (4) | (5) | (6) |
|---|---|---|---|---|---|---|
| **Influencing Mechanism** | **Technical Progress Effect** | | **Scale Expansion Effect** | | **Innovation-Driven Effect** | |
| **Variable** | **inno** | **GHQ** | **gf** | **GHQ** | **iur** | **GHQ** |
| DID | 4.1862 ** | 0.0491 * | 0.0092 * | 0.0375 ** | 0.7752 *** | 0.0454 * |
| | (1.6197) | (0.0241) | (0.0064) | (0.0157) | (0.2029) | (0.0229) |
| inno | | 0.0007 | | | | |
| | | (0.0045) | | | | |
| gf | | | | 0.2352 | | |
| | | | | (0.2968) | | |
| iur | | | | | | 0.0086 ** |
| | | | | | | (0.0036) |
| lnsecond | 2.3883 | −0.0499 | 0.0296 ** | −0.0614 *** | 0.9313 | −0.0563 |
| | (1.6923) | (0.0344) | (0.0124) | (0.0150) | (0.6431) | (0.0329) |
| lnthrid | −0.2417 | 0.1635 *** | 0.0106 | 0.0881 *** | 1.7420 * | 0.1484 *** |
| | (3.8156) | (0.0423) | (0.0197) | (0.0296) | (0.8813) | (0.0414) |
| GS | −6.0367 | 0.1830 | 0.0161 | 0.0674 | −0.9925 | 0.1872 |
| | (16.2767) | (0.2461) | (0.0773) | (0.1992) | (2.8196) | (0.2311) |
| FD | −0.3385 | 0.0499 ** | −0.0093 * | 0.0207 * | 0.8333 ** | 0.0425 * |
| | (1.4870) | (0.0227) | (0.0057) | (0.0114) | (0.3689) | (0.0232) |
| ETR | 3.2291 * | −0.0110 | 0.0020 | −0.0140 ** | 0.1372 | −0.0099 |
| | (1.6082) | (0.0201) | (0.0027) | (0.0053) | (0.3335) | (0.0116) |
| EC | 8.3649 *** | −0.0533 | 0.0120 | −0.0504 * | 0.2974 | −0.0500 * |
| | (1.9971) | (0.0551) | (0.0120) | (0.0244) | (0.4009) | (0.0250) |
| Constant | −20.4242 | −0.9452 * | −0.1972 * | −0.1023 | −14.9113 | −0.8314 * |
| | (44.2078) | (0.4554) | (0.1090) | (0.1870) | (12.2870) | (0.4483) |
| $R^2$ | 0.9573 | 0.7034 | 0.9807 | 0.7072 | 0.8561 | 0.7055 |
| N | 119 | 119 | 119 | 119 | 119 | 119 |
| Year-fixed effect | Yes | Yes | Yes | Yes | Yes | Yes |
| Province-fixed effect | Yes | Yes | Yes | Yes | Yes | Yes |

Note: *, **, and *** indicate that the statistical value is significant at 10%, 5%, and 1%, respectively. Values in parentheses are standard deviations.

**Table 9.** Sobel test and Bootstrap test results of influencing mechanism.

| | | Technical Progress Effect | Scale Expansion Effect | Innovation-Driven Effect |
|---|---|---|---|---|
| | Meditating effect | 7.38 | 5.45 | 16.79 |
| Sobel test | Std Err | 0.0044 | 0.0045 | 0.0017 |
| | Z | 6.87 *** | 6.682 *** | 2.284 *** |
| Bootstrap test | [95% conf. interval] | [0.0183, 0.0434] | [0.0208, 0.0392] | [0.0047, 0.0073] |

Note: *** indicates that the statistical value is significant at 1%.

### 5.2. Scale Expansion Mediating Effects

Economic scale expansion is a natural outcome of PFTZ implementation. This paper draws on the study of [45] and uses the GHD Index to measure the scale expansion capability of the regional economy. From column (3) of Table 8, it can be seen that the coefficient of the effect of PFTZs on scale expansion is significantly positive. This indicates that PFTZs effectively promote the development of green finance and expand the size of the regional economy. The results in column (4) are significant, and the regression coefficient is significantly positive at the 5% level after including the scale expansion in the model, while the regression coefficient of green finance is not significant. This research examines the influence mechanism of scale expansion using the same procedure as above. The findings of Table 9 demonstrate that the scale expansion Sobel test's *p*-value is 0.0045, which is less

than 0.05. The confidence interval of the Bootstrap test is [0.0208, 0.0392], which does not contain 0. The results indicate that the scale expansion effect has a mediating influence. It can be stated that the scale expansion effect partially mediates the process by which PFTZs enhance regional green and high-quality development. Thus, H2b is supported.

*5.3. Innovation-Driven Mediating Effects*

Innovation-driven is an essential way for PFTZs to play their policy role. In this paper, industry–university–research cooperation is defined in terms of the R&D expenditure of enterprises that support higher education and research and development institutions. The higher the degree of industry–university–research cooperation, the higher the innovation openness and the more favorable the innovation environment [46]. From column (5), it can be seen that the coefficient of the effect of PFTZs on industry–university–research cooperation is significantly positive. This indicates that PFTZs effectively promote cooperation between industry, university, and research and form the innovation drive. The results in column (6) show that the estimated coefficients of the dummy variable DID remain significant after the inclusion of the innovation-driven model, and the mediating effect is significant. Additionally, the results in Table 9 confirm the robustness of the innovation-driven influencing mechanism. The *p*-value of the Sobel test is 0.0017, which is less than 0.05. The confidence interval of the Bootstrap test is [0.0047, 0.0073], which does not contain 0. This indicates that the innovation-driven effect plays a partially mediating role in the process of PFTZs for green and high-quality development. H2c is proven.

## 6. Conclusions and Policy Insights

As the link between domestic and foreign commerce, PFTZs act as a driver for internal and external circulation and a crucial foundation for the advancement of high-quality green trade. Therefore, the construction of PFTZs is crucial to transform the economic development model and promote green and high-quality development. This study uses PSM-DID to evaluate the policy effects and underlying processes of PFTZ building on green and high-quality development based on panel data of Chinese provinces from 2007 to 2020. The results show that (1) the construction of PFTZs significantly enhances regional green and high-quality development, with a net policy impact of roughly 4%. The several robustness tests demonstrate that the outcomes are extremely reliable. (2) According to the heterogeneity analysis, compared with inland areas, coastal locations may more effectively foster green and high-quality development. (3) The influencing mechanism test results show that PFTZs primarily have a positive impact on technological progress, scale expansion, and innovation-driven effects, which, in turn, support green and high-quality development. The mediating effect of innovation-driven and technological progress is extremely significant among them.

Based on the above conclusions, several policy insights for the sustainable growth of PFTZs might be provided.

First and foremost, the government and management of the relevant PFTZs should keep up their support for the development of new free trade zones, maintain a deep and high level of openness, and facilitate the flow of high-quality factors and resources. We should pay more attention to improving the level and quality of PFTZs and, thus, drive the green and high-quality development of the local economy, while we gradually expand the number of provincial free trade zones. We should focus more on promoting the local economy's development of high standards and sustainability. Second, by determining the PFTZs' functional placement, local businesses can be encouraged to avoid low-end industrialization, gradually eliminate environmentally polluting industries, and support the transition of the industrial structure to one that is more environmentally friendly. Inland FTZs should also lessen the tax burden on businesses. Finally, according to the research in this paper, technological progress and green finance do not play a sufficient role in promoting green and high-quality development at this stage. PFTZs should, therefore, take note of technical innovation and breakthroughs. Furthermore, to improve the sustainability

of regional economic development, PFTZs should create a comprehensive green finance system with green-oriented features.

This study recognizes the crucial function of PFTZs in the new pattern of development and establishes that PFTZ construction can effectively promote regional green and high-quality development. In other words, PFTZs can boost environmental advantages while enhancing the quality and effectiveness of factor flows. The research findings provide targeted policy recommendations for China's PFTZs to support green and high-quality development. This also serves as a foundation for decision-making for the global development of PFTZs. There are still certain limitations, nevertheless, that need further development. We will analyze PFTZs and the countries along the Belt and Road in the future and thoroughly explore how China's PFTZs affect the green and high-quality development of other nations in light of the parallels between the two policies. Additionally, the spatial spillover effect on PFTZ building is a question worthy of more research.

**Author Contributions:** Conceptualization, L.L. and Y.C.; Methodology, L.L.; Software, L.L.; Validation, Y.C.; Formal Analysis, Y.C. and L.L.; Data Curation, Y.C. and L.L.; Writing—Original Draft Preparation, L.L.; Writing—Review and Editing, L.L. and Y.C.; Visualization, L.L.; Supervision, Y.C.; Project Administration, Y.C.; Funding Acquisition, Y.C. All authors have read and agreed to the published version of the manuscript.

**Funding:** This research was funded by the National Natural Science Foundation of China (72104185), the Shanghai Sailing Program (21YF1449600), and the Philosophy and Social Science Foundation of Shanghai (2019BDS005).

**Institutional Review Board Statement:** Not applicable.

**Informed Consent Statement:** Not applicable.

**Data Availability Statement:** The data used in this study are available from the corresponding author on reasonable request.

**Conflicts of Interest:** The authors declare no conflict of interest.

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
