# Peer review of "Effects and Mechanisms of China’s Pilot Free Trade Zones on Green and High-Quality Development from the Dual-Circulation Perspective"

_sustainability, doi:10.3390/su15020947_

Round 1

Reviewer 1 Report

This paper provides some interesting insights on the pivotal role of China’s PFTZs in the new development pattern, and it is of great significance in promoting GDC. The discussion on the relationship between China’s PFTZs and GDC, especially the discussion on the policy linkage effect, is of some practical value. However, I consider that many of the main points in this paper are explained very vaguely. For this reason, in my view it needs the following modifications:

1. The concepts of GDC are not clearly defined throughout the paper. The authors only mentioned that " The goal of green dual-circulation development (GDC) is to achieve high-quality economic development while adhering to the double-cycle principle. It is able to fully utilize both home and international markets as well as the two different types of resources, realizing the domestic and international double cycle for mutual promotion". But that doesn't explain what GDC is or how it differs from dual-circulation.

2. The research contributions summarized in this paper are not convincing. In particular, the robustness test and heterogeneity test do not reflect the innovation of this paper. Therefore, in my view this paper lacks innovation and novelty.

3. The comprehensive index system of the GDC level from the two dimensions of internal and external circulation is unreasonable and unconvincing. These indicators fail to measure the level of environmental development of a province, but only show the level of economic development. Thus, the explained variables are almost irrelative to GDC. GDC should include more environmental indicators, such as CO2 emissions, atmospheric pollution, or green patents. More importantly, GDC is more about "circulation", not just levels of development.

Author Response

The reply is uploaded as an attachment, please see the attachment for details.

Point 1: The concepts of GDC are not clearly defined throughout the paper. The authors only mentioned that " The goal of green dual-circulation development (GDC) is to achieve high-quality economic development while adhering to the double-cycle principle. It is able to fully utilize both home and international markets as well as the two different types of resources, realizing the domestic and international double cycle for mutual promotion". But that doesn't explain what GDC is or how it differs from dual-circulation.

Response 1: Thank you for the comment and some corrections are made.

Green dual-circulation development (GDC) is a green, low-carbon, sustainable, and high-quality development model that is founded on the principle of ecological priority and creates a dynamic balance between domestic and international markets. Compared with dual circulation, green dual-circulation places more emphasis on green and high-quality development. However, according to response 3, we modify the indicator system by adding the level of economic development and environmental factors. Therefore, we sublimate green double-circulation to green high-quality development based on the revision of the index system and the connotation of GDC.

Point 2: The research contributions summarized in this paper are not convincing. In particular, the robustness test and heterogeneity test do not reflect the innovation of this paper. Therefore, in my view this paper lacks innovation and novelty.

Response 2: Thank you for the comment and some corrections are made.

According to your advice, we rework the research contribution and innovativeness. Meanwhile, the study method is improved. DID method can be used to estimate policy effects, however its estimation findings may contain biased data. Therefore, we employ the PSM-DID method for policy evaluation to get over this issue. The specific modifications are as follows.

“In light of this, this paper considering PFTZs as a quasi-natural experiment, utilizes the propensity score matching and difference in difference method (PSM-DID) to evaluate the policy effects of PFTZs construction on green and high-quality development. The main research contributions of this paper include the following. (1) Considering the key linkage role of PFTZs in the new development pattern of dual circulation, this paper constructs a comprehensive evaluation system of green and high-quality development in four aspects, including internal and external circulation, economic development and environment, so as to quantify the quality of green development in each region. Additionally, in order to investigate the causal connection between PFTZs and green high-quality development, this paper incorporates both into the same research framework. (2) Current studies mostly evaluate the design of PFTZs using the twofold difference approach, but they are unable to address the issue of selective bias in the estimated sample. This paper uses PSM-DID for regression analysis to effectively eliminate the influence of unobservable factors and thus obtain the true net effect of policy implementation. (3) The influence of PFTZs on environmental consequences is investigated individually at the level of the mechanism in comparison to earlier work. This paper explains the internal driving paths of the policy effects and theoretically and empirically investigates the various implications of PFTZs on sustainable and high-quality development.” which can be seen in Lines 110-127.

We add the propensity score matching (PSM) method to the original model (multi-period DID). Based on the propensity score results, the net effect generated by the policy is tested using multi-period DID. Additionally, we modify the original model formulation.

Point 3: The research contributions summarized in this paper are not convincing. In particular, the robustness test and heterogeneity test do not reflect the innovation of this paper. Therefore, in my view this paper lacks innovation and novelty.

Response 3: Thank you for the comment and some corrections are made.

According to your advice, based on the original index system, we consider the current status of regional basic economic development, environmental regulation and pollution. Provided that data are available, we add two dimensions of economic development and environmental factors with six sub-indicators. The specific modifications are as follows.

“Green high-quality development is the explained variable. With the faster implementation of the new growth pattern, China’s economic and social development has given more attention to the quality and efficiency of green factor flows than most previous research on green high-quality development achieved [32, 34-36]. Therefore, this paper draws on the study of Wang et al. [37]to construct a green and high-quality development index (GHD) to represent the quality of regional green development by using the entropy weighting method. Table 1 shows the description and weights of relevant indicators.” which can be seen in Lines 223-229.

Besides the above revisions, we do the following revisions.

First, we revise the title and abstract. The specific modifications are as follows.

Title: “Do China’s PFTZ promote green and high-quality development from the dual-circulation perspective”

Abstract: “Under the dual pressure of economic growth and environmental pressure, green and high-quality development become a critical strategy for China to achieve win-win development for both the economy and the environment. In the dual-circulation pattern, a deep and high level of openness is the primary prerequisite for encouraging green and high-quality development. Previous studies have overlooked the crucial pivotal role of PFTZs in the domestic and international circulation. The PFTZ is essential in driving China’s green modernization and high-quality development as a crucial area of opening up. This paper utilizes the propensity score matching and difference in difference (PSM-DID) on panel data from 27provinces from 2007 to 2020 to investigate the green policy effects of PFTZs building and its underlying mechanism. The results show that (1) the construction of PFTZs significantly enhance regional green and high-quality development, with a net policy impact of roughly 4%. The several robustness tests demonstrate that the outcomes are extremely reliable. (2) According to the heterogeneity analysis, compared with inland areas, coastal locations may more effectively foster green and high-quality development. (3) The influencing mechanism test results show that PFTZs primarily have positive impact on technological progress, scale expansion, and innovation-driven effects, which in turn support green and high-quality development. The mediating effect of innovation-driven and technological progress is extremely significant among them. These research findings can provide empirical support and policy rationale for PFTZs to better promote China’s green and high-quality development.”

Second, to make the introduction easier for the reader to understand, we revise it simultaneously. Since we add the theoretical analysis of the influencing mechanisms, we organize the literature review on PFTZs into the introduction section.

Third, we re-perform the empirical analysis due to the modification of the indicator system and research methodology. As a result, the empirical analysis section is significantly revised. For details, please see Section 4.

Fourth, we revise the conclusion section. The following are the specific revisions.

“As the link between domestic and foreign commerce, PFTZs acts as a driver for internal and external circulation and a crucial foundation for the advancement of high-quality green trade. Therefore, the construction of PFTZs is crucial to transform the economic development model and promote green and high-quality development. This study uses PSM-DID to evaluate the policy effects and underlying processes of PFTZs building on green and high-quality development based on panel data of Chinese provinces from 2007 to 2020. The results show that (1) the construction of PFTZs significantly enhance regional green and high-quality development, with a net policy impact of roughly 4%. The several robustness tests demonstrate that the outcomes are extremely reliable. (2) According to the heterogeneity analysis, compared with inland areas, coastal locations may more effectively foster green and high-quality develop-ment. (3) The influencing mechanism test results show that PFTZs primarily have positive impact on technological progress, scale expansion, and innovation-driven effects, which in turn support green and high-quality development. The mediating effect of innovation-driven and technological progress is extremely significant among them.

Based on the above conclusions, several policy insights for the sustainable growth of PFTZs might be provided.

First and foremost, the government and management of the relevant PFTZs should keep up their support for the development of new free trade zones, maintain a deep and high level of openness, and facilitate the flow of high-quality factors and resources. We should pay more attention to improving the level and quality of PFTZs and thus driving the green and high-quality development of the local economy, while we gradually expand the number of provincial free trade zones. we should focus more on promoting the local economy’s development of high standards and sustainability. Second, by determining the PFTZs’ functional placement, local businesses can be encouraged to avoid low-end industrialization, gradually eliminate environmentally polluting industries, and support the transition of the industrial structure to one that is more environmentally friendly. Inland PFTZs should also lessen the tax burden on businesses. Finally, according to the research in this paper, technological progress and green finance do not play a sufficient role in promoting green and high-quality development at this stage. PFTZs should therefore take note of technical innovation and breakthroughs. Furthermore, to improve the sustainability of regional economic development, PFTZs should create a comprehensive green finance system with green-oriented features.

This study recognizes the crucial function of PFTZs in the new pattern of devel-opment and establishes that PFTZs construction can effectively promote regional green and high-quality development. In other words, PFTZs can boost environmental advantages while enhancing the quality and effectiveness of factor flows. The research findings provide targeted policy recommendations for China’s PFTZs to support green and high-quality development. It also serves as a foundation for decision-making for the global development of PFTZs. There are still certain limitations, nevertheless, that need further development. We will analyze PFTZs and the countries along the Belt and Road in the future and thoroughly explore how China's PFTZs affect the green and high-quality development of other nations in light of the parallels between the two policies. Additionally, the spatial spillover effect on PFTZ building is a question worthy of more research.” which can be seen in Lines 413-456.

Lastly, more relevant papers and books are cited in the manuscript and are included in the References, which are listed below.

  1. BP statistical review of world energy 2022. London: British Petroleum Public Limited Company; 2022.
  2. Zhou G, Zhang Z, Fei Y. How to Evaluate the Green and High-Quality Development Path? An FsQCA Approach on the China Pilot Free Trade Zone. International Journal of Environmental Research and Public Health. 2022;19(1).
  3. Ye P, Zhang H, Ma S, Yang F, Li Y. A Knowledge Map Study of an Application of a Smart Land Planning Free-Trade Zone and China’s Contribution. Land. 2022;11(6).
  4. Zhao T, He F. Does the Pilot Free Trade Zone Promote the Quality of Urban Economic Growth: An Empirical Research Based on Quasi-Natural Experiment. Sustainability. 2022;14(12).
  5. Alvarez-Villa D, Guardado J. The long-run influence of institutions governing trade: Evidence from smuggling ports in co-lonial Mexico. Journal of Development Economics. 2020;144.
  6. Yang S, Martinez-Zarzoso I. A panel data analysis of trade creation and trade diversion effects: The case of ASEAN–China Free Trade Area. China Economic Review. 2014;29:138-51.
  7. Yanase A, Tsubuku M. Trade costs and free trade agreements: Implications for tariff complementarity and welfare. Interna-tional Review of Economics & Finance. 2022;78:23-37.
  8. Tachie AK, Xingle L, Dauda L, Mensah CN, Appiah-Twum F, Adjei Mensah I. The influence of trade openness on envi-ronmental pollution in EU-18 countries. Environmental Science and Pollution Research International. 2020;27(28):35535-55.
  9. He J. Pollution haven hypothesis and environmental impacts of foreign direct investment: The case of industrial emission of sulfur dioxide (SO) in Chinese provinces2. Ecological Economics. 2006;60(1):228-45.
  10. Wang L, Wang Z, Ma Y. Environmental Effects of Trade Liberalization–Evidence from China’s Pilot Free Trade Zone. Journal of International Commerce, Economics and Policy. 2021;12(03).
  11. Wang H, Zhang Y, Liu Z, Liu R, Li K. The impact and mechanisms of the Shanghai pilot free-trade zone on the green total factor productivity of the Yangtze River Delta Urban Agglomeration. Environmental Science and Pollution Research International. 2022;29(27):40997-1011.
  12. Erdogan AM. Bilateral trade and the environment: a general equilibrium model based on new trade theory. International Review of Economics & Finance. 2014;34:52-71.
  13. Li X, Huang Y, Li X, Liu X, Li J, He J, et al. How does the Belt and Road policy affect the level of green development? A quasi-natural experimental study considering the CO2 emission intensity of construction enterprises. Humanities and Social Sciences Communications. 2022;9(1):1-11.
  14. Zhang E, He X, Xiao P. Does Smart City Construction Decrease Urban Carbon Emission Intensity? Evidence from a Differ-ence-in-Difference Estimation in China. Sustainability. 2022;14(23).
  15. Wang S, Gao S, Huang Y, Shi C. Spatiotemporal evolution of urban carbon emission performance in China and prediction of future trends. Journal of Geographical Sciences. 2020;30:757–74.
  16. Grossman GM, Krueger AB. Economic Growth and the Environment. The Quarterly Journal of Economics. 1995;110(2):353-77.
  17. Fei R, Cui A, Qin K. Can technology R&D continuously improve green development level in the open economy? Empirical evidence from China’s industrial sector. Environmental Science and Pollution Research. 2020;27:34052–66.
  18. Anser MK, Khan MA, Nassani AA, Aldakhil AM, Voo XH, Zaman K. Relationship of environment with technological in-novation, carbon pricing, renewable energy, and global food production. Economics of Innovation and New Technology. 2021;30(8):807-42.
  19. An T, Xu C, Liao X. The impact of FDI on environmental pollution in China: evidence from spatial panel data. Environmental Science and Pollution Research International. 2021;28(32):44085-97.
  20. Lee C-C, Wang F. How does digital inclusive finance affect carbon intensity? Economic Analysis and Policy. 2022;75:174-90.
  21. Lin B, Ma R. Green technology innovations, urban innovation environment and CO emission reduction in China: Fresh ev-idence from a partially linear functional-coefficient panel model2. Technological Forecasting and Social Change. 2022;176.
  22. Rosenbaum PR, Rubin DB. The Central Role of the Propensity Score in Observational Studies For Causal Effects. Biometrika. 1983;70(1):41-55.
  23. Xin B, Qu Y. Effects of Smart City Policies on Green Total Factor Productivity: Evidence from a Quasi-Natural Experiment in China. International Journal  of Environmental Research and Public Health. 2019;16(13).
  24. Beck T, Levine R, Levkov A. Big Bad Banks? The Winners and Losers from Bank Deregulation in the United States. Journal of Finance. 2010;65(5):1637-67.
  25. Wu H, Hao Y, Ren S. How do environmental regulation and environmental decentralization affect green total factor energy efficiency: Evidence from China. Energy Economics. 2020;91(C).
  26. Abbasab J, Sağsana M. Impact of knowledge management practices on green innovation and corporate sustainable devel-opment: A structural analysis. Journal of Cleaner Production. 2019;229:611-20.
  27. Fraccascia L, Giannoccaro I, Albino V. Green product development: What does the country product space imply? Journal of Cleaner Production. 2018;170:1076-88.
  28. Wang F, Wang R, He Z. Exploring the Impact of “Double Cycle” and Industrial Upgrading on Sustainable High-Quality Economic Development: Application of Spatial and Mediation Models. Sustainability. 2022;14(4).
  29. Baron RM, Kenny DA. The moderator-mediator variable distinction in social psychological research: conceptual, strategic, and statistical considerations. Journal of Personality and Social Psychology. 1986;51:1173-82.
  30. Acemoglu D, Akcigit U, Kerr W. Innovation network. Proceedings of the National Academy of Sciences of the United States of America. 2016; 113(41):11483-8.
  31. Anser MK, Khan MA, Awan U, Batool R, Zaman K, Imran M, et al. The Role of Technological Innovation in a Dynamic Model of the Environmental Supply Chain Curve: Evidence from a Panel of 102 Countries. Processes. 2020;8(9).
  32. He L, Liu R, Zhong Z, Wang D, Xia Y. Can green financial development promote renewable energy investment efficiency? A consideration of bank credit. Renewable Energy. 2019;143:974-84.
  33. Cui W, Li L, Chen G. Market-value oriented or technology-value oriented? ——Location impacts of indus-try-university-research (IUR) cooperation bases on innovation performance. Technology in Society. 2022;70.

Reviewer 2 Report

1. In the abstract part of the manuscript, 1-2 sentences should be used to describe the research background and the limitations of previous studies.
2. Please point out the research results in the abstract to make this part more clear.
3. The inspiration and significance of the manuscript should be added after the abstract.
4. In Introduction section, "In addition, it also has strong and practical guidance significance" In the manuscript only indicates that the manuscript is of significance. Please give a detailed description of the significance of the study.
5. In Introduction section, the research content of the introduction part of the manuscript is suggested to be divided into separate paragraphs.
6. In Introduction section, please propose the scientific problems that need to be solved in this article in the form of question sentences.
7. In Literature review section, please give a specific explanation of the technical term "the pilot free trade zone" in the manuscript.
8. In addition, some important studies using Quasi-Natural Experimental methods in the field of green development are missed.
e.g.,
Li, X., Huang, Y., Li, X., Liu, X., Li, J., He, J., & Dai, J. (2022). How does the Belt and Road policy affect the level of green development? A quasi-natural experimental study considering the CO2 emission intensity of construction enterprises. Humanities and Social Sciences Communications, 9(1), 1-11.
Luo, S., Yu, S., & Zhou, G. (2021). Does green credit improve the core competence of commercial banks? Based on quasi-natural experiments in China. Energy Economics, 100, 105335.
Qiu, S., Wang, Z., & Liu, S. (2021). The policy outcomes of low-carbon city construction on urban green development: Evidence from a quasi-natural experiment conducted in China. Sustainable Cities and Society, 66, 102699.
Xin, B., & Qu, Y. (2019). Effects of smart city policies on green total factor productivity: evidence from a quasi-natural experiment in China. International journal of environmental research and public health, 16(13), 2396.
etc.
9. In Methodology and data section, plsease explain the advantages of this method, its applicability, and the reasons for not using other methods, and the author is invited to add instructions.
10. In Methodology and data section, please add the appropriate comment to the data source.
11. In Empirical results section, “employees in six industries, including scientific research,education, finance, culture and sports, information transmission, and leasing services,were defined as innovative talent.”the manuscript defining Whether there is a basis for defining the practitioners of these 6 industries as innovative talents.
12. In Empirical results section, Fig 2 in the manuscript text lacks a specific description of the line, after reading can not distinguish which line represents what, ask the author to modify the diagram for optimization.
13. Please add a discussion section to discuss whether other similar studies are consistent with the results of this study and, if so, to analyze the causes.
14. In conclusion section, it should describe the research purpose, research content, method and general description of the manuscript in 1-3 sentences. The author is invited to add this part.
15. In conclusion section, it is recommended to list the conclusions of the manuscript in points.
16. In conclusion section, please make targeted policy recommendations based on the results of this study.

To sum up, it is suggested that the authors revise the manuscript in the light of the above suggestions. I sincerely look forward to receiving the revised version.

Author Response

The reply is uploaded as an attachment, please see the attachment for details.

Point 1 - About Abstrct: (1) In the abstract part of the manuscript, 1-2 sentences should be used to describe the research background and the limitations of previous studies. (2) Please point out the research results in the abstract to make this part more clear. The inspiration and significance of the manuscript should be added after the abstract.

Response 1: Thank you for the comment and some corrections are made.

According to your advice, we revise the abstract. The specific modifications are as follows.

Abstract: “Under the dual pressure of economic growth and environmental pressure, green and high-quality development become a critical strategy for China to achieve win-win development for both the economy and the environment. In the dual-circulation pattern, a deep and high level of openness is the primary prerequisite for encouraging green and high-quality development. Previous studies have overlooked the crucial pivotal role of PFTZs in the domestic and international circulation. The PFTZ is essential in driving China’s green modernization and high-quality development as a crucial area of opening up. This paper utilizes the propensity score matching and difference in difference (PSM-DID) on panel data from 27provinces from 2007 to 2020 to investigate the green policy effects of PFTZs building and its underlying mechanism. The results show that (1) the construction of PFTZs significantly enhance regional green and high-quality development, with a net policy impact of roughly 4%. The several robustness tests demonstrate that the outcomes are extremely reliable. (2) According to the heterogeneity analysis, compared with inland areas, coastal locations may more effectively foster green and high-quality development. (3) The influencing mechanism test results show that PFTZs primarily have positive impact on technological progress, scale expansion, and innovation-driven effects, which in turn support green and high-quality development. The mediating effect of innovation-driven and technological progress is extremely significant among them. These research findings can provide empirical support and policy rationale for PFTZs to better promote China’s green and high-quality development.”

Point 2 - About Introduction: (1) In Introduction section, "In addition, it also has strong and practical guidance significance" In the manuscript only indicates that the manuscript is of significance. Please give a detailed description of the significance of the study. (2) In Introduction section, the research content of the introduction part of the manuscript is suggested to be divided into separate paragraphs. (3) In Introduction section, please propose the scientific problems that need to be solved in this article in the form of question sentences.

Response 2: Thank you for the comment and some corrections are made.

According to your advice, we revise the Introduction. To make the introduction easier for the reader to understand, we revise it simultaneously. Since we add the theoretical analysis of the influencing mechanisms, we organize the literature review on PFTZs into the introduction section. 

Point 3 – About Method : (1) In Methodology and data section, plsease explain the advantages of this method, its applicability, and the reasons for not using other methods, and the author is invited to add instructions. (2) In Methodology and data section, please add the appropriate comment to the data source.

Response 3: Thank you for the comment and some corrections are made.

According to your advice, we revise the research methods section. Meanwhile, the original model has the issue with selective bias. Therefore, we employ the PSM-DID method for policy evaluation to get over this issue.  Additionally, we modify the original model formulation. 

Point 4 – About Empirical results: (1) In Empirical results section, “employees in six industries, including scientific research, education, finance, culture and sports, information transmission, and leasing services, were defined as innovative talent.” the manuscript defining Whether there is a basis for defining the practitioners of these 6 industries as innovative talents. (2) In Empirical results section, Fig 2 in the manuscript text lacks a specific description of the line, after reading can not distinguish which line represents what, ask the author to modify the diagram for optimization.

Response 4: Thank you for the comment and some corrections are made.

The originally selected mechanism variable-innovative talent is an indicator selected with reference to the study of Wang et al. (Wang M, Xuan Y, Chen Q. Creative Class Agglomeration, Knowledge Externalities and Urban Innovation — Evidence from 20 Large Cities. Economic Theory and Business Management. 2016; (01):59-70.)

Then, we restructure the variables for the empirical analysis due to the revision of the indicator system. Additionally, we describe specifically the parallel trend test (Figure 3). The specific modifications are as follows.

First, based on the original index system, we consider the current status of regional basic economic development, environmental regulation and pollution. Provided that data are available, we add two dimensions of economic development and environmental factors with six sub-indicators. Furthermore, we sublimate green double-circulation to green high-quality development based on the revision of the index system and the connotation of green dual circulation.

Second, we search for more suitable control variables as a result of the indicator system and study methodological changes. 

Third, we optimize the research hypothesis. The original hypothesis lacks theoretical support. After modification, we analyze the influencing mechanism of PFTZs on green and high-quality development based on EKC theory. Therefore, technological progress, scale expansion and innovation drive are chosen as mediating paths. 

Additionally, we reformulate the mediation model. 

Point 5 – About Conclusion: (1) In conclusion section, it should describe the research purpose, research content, method and general description of the manuscript in 1-3 sentences. The author is invited to add this part. (2) In conclusion section, it is recommended to list the conclusions of the manuscript in points. (3) In conclusion section, please make targeted policy recommendations based on the results of this study.

Response 5: Thank you for the comment and some corrections are made.

According to your advice, we revise the conclusion section. 

The following are the specific revisions.

“As the link between domestic and foreign commerce, PFTZs acts as a driver for internal and external circulation and a crucial foundation for the advancement of high-quality green trade. Therefore, the construction of PFTZs is crucial to transform the economic development model and promote green and high-quality development. This study uses PSM-DID to evaluate the policy effects and underlying processes of PFTZs building on green and high-quality development based on panel data of Chinese provinces from 2007 to 2020. The results show that (1) the construction of PFTZs significantly enhance regional green and high-quality development, with a net policy impact of roughly 4%. The several robustness tests demonstrate that the outcomes are extremely reliable. (2) According to the heterogeneity analysis, compared with inland areas, coastal locations may more effectively foster green and high-quality development. (3) The influencing mechanism test results show that PFTZs primarily have positive impact on technological progress, scale expansion, and innovation-driven effects, which in turn support green and high-quality development. The mediating effect of innovation-driven and technological progress is extremely significant among them.

Based on the above conclusions, several policy insights for the sustainable growth of PFTZs might be provided.

First and foremost, the government and management of the relevant PFTZs should keep up their support for the development of new free trade zones, maintain a deep and high level of openness, and facilitate the flow of high-quality factors and re-sources. We should pay more attention to improving the level and quality of PFTZs and thus driving the green and high-quality development of the local economy, while we gradually expand the number of provincial free trade zones. we should focus more on promoting the local economy’s development of high standards and sustainability. Second, by determining the PFTZs’ functional placement, local businesses can be encouraged to avoid low-end industrialization, gradually eliminate environmentally polluting industries, and support the transition of the industrial structure to one that is more environmentally friendly. Inland PFTZs should also lessen the tax burden on businesses. Finally, according to the research in this paper, technological progress and green finance do not play a sufficient role in promoting green and high-quality development at this stage. PFTZs should therefore take note of technical innovation and breakthroughs. Furthermore, to improve the sustainability of regional economic development, PFTZs should create a comprehensive green finance system with green-oriented features.

This study recognizes the crucial function of PFTZs in the new pattern of development and establishes that PFTZs construction can effectively promote regional green and high-quality development. In other words, PFTZs can boost environmental advantages while enhancing the quality and effectiveness of factor flows. The research findings provide targeted policy recommendations for China’s PFTZs to support green and high-quality development. It also serves as a foundation for decision-making for the global development of PFTZs. There are still certain limitations, nevertheless, that need further development. We will analyze PFTZs and the countries along the Belt and Road in the future and thoroughly explore how China's PFTZs affect the green and high-quality development of other nations in light of the parallels between the two policies. Additionally, the spatial spillover effect on PFTZ building is a question worthy of more research.” which can be seen in Lines 413-456.

Besides the above revisions, we do the following revisions.

First, we revise the title to reflect the redefinition of the explained variables.

Title: “Do China’s PFTZ promote green and high-quality development from the dual-circulation perspective”

Second, we re-perform the empirical analysis due to the modification of the indicator system and research methodology. As a result, the empirical analysis section is significantly revised. For details, please see Section 4.

Third, more relevant papers and books are cited in the manuscript and are included in the References, which are listed below.

  1. BP statistical review of world energy 2022. London: British Petroleum Public Limited Company; 2022.
  2. Zhou G, Zhang Z, Fei Y. How to Evaluate the Green and High-Quality Development Path? An FsQCA Approach on the China Pilot Free Trade Zone. International Journal of Environmental Research and Public Health. 2022;19(1).
  3. Ye P, Zhang H, Ma S, Yang F, Li Y. A Knowledge Map Study of an Application of a Smart Land Planning Free-Trade Zone and China’s Contribution. Land. 2022;11(6).
  4. Zhao T, He F. Does the Pilot Free Trade Zone Promote the Quality of Urban Economic Growth: An Empirical Research Based on Quasi-Natural Experiment. Sustainability. 2022;14(12).
  5. Alvarez-Villa D, Guardado J. The long-run influence of institutions governing trade: Evidence from smuggling ports in co-lonial Mexico. Journal of Development Economics. 2020;144.
  6. Yang S, Martinez-Zarzoso I. A panel data analysis of trade creation and trade diversion effects: The case of ASEAN–China Free Trade Area. China Economic Review. 2014;29:138-51.
  7. Yanase A, Tsubuku M. Trade costs and free trade agreements: Implications for tariff complementarity and welfare. International Review of Economics & Finance. 2022;78:23-37.
  8. Tachie AK, Xingle L, Dauda L, Mensah CN, Appiah-Twum F, Adjei Mensah I. The influence of trade openness on environmental pollution in EU-18 countries. Environmental Science and Pollution Research International. 2020;27(28):35535-55.
  9. He J. Pollution haven hypothesis and environmental impacts of foreign direct investment: The case of industrial emission of sulfur dioxide (SO) in Chinese provinces2. Ecological Economics. 2006;60(1):228-45.
  10. Wang L, Wang Z, Ma Y. Environmental Effects of Trade Liberalization–Evidence from China’s Pilot Free Trade Zone. Journal of International Commerce, Economics and Policy. 2021;12(03).
  11. Wang H, Zhang Y, Liu Z, Liu R, Li K. The impact and mechanisms of the Shanghai pilot free-trade zone on the green total factor productivity of the Yangtze River Delta Urban Agglomeration. Environmental Science and Pollution Research International. 2022;29(27):40997-1011.
  12. Erdogan AM. Bilateral trade and the environment: a general equilibrium model based on new trade theory. International Review of Economics & Finance. 2014;34:52-71.
  13. Li X, Huang Y, Li X, Liu X, Li J, He J, et al. How does the Belt and Road policy affect the level of green development? A quasi-natural experimental study considering the CO2 emission intensity of construction enterprises. Humanities and Social Sciences Communications. 2022;9(1):1-11.
  14. Zhang E, He X, Xiao P. Does Smart City Construction Decrease Urban Carbon Emission Intensity? Evidence from a Difference-in-Difference Estimation in China. Sustainability. 2022;14(23).
  15. Wang S, Gao S, Huang Y, Shi C. Spatiotemporal evolution of urban carbon emission performance in China and prediction of future trends. Journal of Geographical Sciences. 2020;30:757–74.
  16. Grossman GM, Krueger AB. Economic Growth and the Environment. The Quarterly Journal of Economics. 1995;110(2):353-77.
  17. Fei R, Cui A, Qin K. Can technology R&D continuously improve green development level in the open economy? Empirical evidence from China’s industrial sector. Environmental Science and Pollution Research. 2020;27:34052–66.
  18. Anser MK, Khan MA, Nassani AA, Aldakhil AM, Voo XH, Zaman K. Relationship of environment with technological in-novation, carbon pricing, renewable energy, and global food production. Economics of Innovation and New Technology. 2021;30(8):807-42.
  19. An T, Xu C, Liao X. The impact of FDI on environmental pollution in China: evidence from spatial panel data. Environmental Science and Pollution Research International. 2021;28(32):44085-97.
  20. Lee C-C, Wang F. How does digital inclusive finance affect carbon intensity? Economic Analysis and Policy. 2022;75:174-90.
  21. Lin B, Ma R. Green technology innovations, urban innovation environment and CO emission reduction in China: Fresh evidence from a partially linear functional-coefficient panel model2. Technological Forecasting and Social Change. 2022;176.
  22. Rosenbaum PR, Rubin DB. The Central Role of the Propensity Score in Observational Studies For Causal Effects. Biometrika. 1983;70(1):41-55.
  23. Xin B, Qu Y. Effects of Smart City Policies on Green Total Factor Productivity: Evidence from a Quasi-Natural Experiment in China. International Journal  of Environmental Research and Public Health. 2019;16(13).
  24. Beck T, Levine R, Levkov A. Big Bad Banks? The Winners and Losers from Bank Deregulation in the United States. Journal of Finance. 2010;65(5):1637-67.
  25. Wu H, Hao Y, Ren S. How do environmental regulation and environmental decentralization affect green total factor energy efficiency: Evidence from China. Energy Economics. 2020;91(C).
  26. Abbasab J, Sağsana M. Impact of knowledge management practices on green innovation and corporate sustainable development: A structural analysis. Journal of Cleaner Production. 2019;229:611-20.
  27. Fraccascia L, Giannoccaro I, Albino V. Green product development: What does the country product space imply? Journal of Cleaner Production. 2018;170:1076-88.
  28. Wang F, Wang R, He Z. Exploring the Impact of “Double Cycle” and Industrial Upgrading on Sustainable High-Quality Economic Development: Application of Spatial and Mediation Models. Sustainability. 2022;14(4).
  29. Baron RM, Kenny DA. The moderator-mediator variable distinction in social psychological research: conceptual, strategic, and statistical considerations. Journal of Personality and Social Psychology. 1986;51:1173-82.
  30. Acemoglu D, Akcigit U, Kerr W. Innovation network. Proceedings of the National Academy of Sciences of the United States of America. 2016; 113(41):11483-8.
  31. Anser MK, Khan MA, Awan U, Batool R, Zaman K, Imran M, et al. The Role of Technological Innovation in a Dynamic Model of the Environmental Supply Chain Curve: Evidence from a Panel of 102 Countries. Processes. 2020;8(9).
  32. He L, Liu R, Zhong Z, Wang D, Xia Y. Can green financial development promote renewable energy investment efficiency? A consideration of bank credit. Renewable Energy. 2019;143:974-84.
  33. Cui W, Li L, Chen G. Market-value oriented or technology-value oriented? ——Location impacts of industry-university-research (IUR) cooperation bases on innovation performance. Technology in Society. 2022;70.

Reviewer 3 Report

#1 According to the connection between "The Belt and Road" shown by the author in Fig. 1 and "Pilot Free Trade Zones Promote Green Dual-Circulation Development" in China's internal circulation, the author ignores too many factors that should be explored in the real situation, and suggests that the author should be more limited on the original topic.

#2 The construction of the research hypothesis lacks an objective source basis. Why is such a causal relationship worth exploring in the case of China, and what are the differences with other countries?

#3 The author lacks relevant secondary data, and the analysis results lack further analysis. Therefore, what kind of comparison between China and other countries can be analyzed and compared.

#4 The overall research is a subjective conclusion, especially in the influence of "The Belt and Road Initiative" in fig 1, and the connection between the research has not been found at all, so there is a big gap between the actual research and the research structure.

Author Response

The reply is uploaded as an attachment, please see the attachment for details.

Point 1: According to the connection between "The Belt and Road" shown by the author in Fig. 1 and "Pilot Free Trade Zones Promote Green Dual-Circulation Development" in China's internal circulation, the author ignores too many factors that should be explored in the real situation, and suggests that the author should be more limited on the original topic.

Response 1: Thank you for the comment and some corrections are made.

According to your advice, we restrict the original topic to address only the policy effects of China’s PFTZs construction on green and high-quality development. In future studies, we will consider the green policy effects of PFTZs on countries along the Belt and Road. This is the limitation of our study.

Point 2: The construction of the research hypothesis lacks an objective source basis. Why is such a causal relationship worth exploring in the case of China, and what are the differences with other countries?

Response 2: Thank you for the comment and some corrections are made.

According to your advice, we add theoretical support to the research hypothesis. The main goal of establishing PFTZs is to provide China with the expertise necessary to create a new open economic system through system innovation. In order to affect green and high-quality development through three mechanisms, this study builds on EKC theory and incorporates the research of related experts. The specific modifications are as follows.

“The main goal of establishing PFTZs is to provide China with the expertise necessary to create a new open economic system through system innovation. Therefore, promoting trade liberalization is the key task for the construction and development of PFTZs. Based on this, the Environment Kuznets Curve (EKC) hypothesis put forward by Grossman and Krueger [25] can be used to explain the theoretical mechanism of PFTZs building on regional green and high-quality development. The primary tenet of this theory is that, in addition to having an effect on domestic economic activities, international trade activities have an impact on the quality of the national environment through scale, structure, and technology effects. In order to affect green and high-quality development through three mechanisms of action—technological progress, scale expansion, and innovation-driven effects—this study builds on EKC theory and incorporates the research of related experts.

Technological progress effect refers to the construction of PFTZs by influencing the level of technology and generating technological change, which in turn affects the quality of green development. By increasing investment in technical R&D and securing original technological achievements, enterprises in the PFTZs can advance technology thanks to the clear, green development-oriented requirements. Fei et al. [26] pointed out that, in an open economy, independent innovation can continuously raise the standard of regional green development. Likewise, Anser et al. [27] demonstrate how the results of technological progress contribute to the achievement of sustainable development goals. Therefore, the establishment of PFTZs can improve the quality of local green economy development through technological progress effect.

The scale expansion effect describes how the building of PFTZs affects the scale of economic activity in the area and, consequently, green high-quality development. Re-lying on institutional advantages, PFTZs provide a liberalized and facilitated policy environment for trade, investment, and financial innovation. This can directly alleviate the difficulties such as lack of capital in the process of local economic development and promote the expansion of the economy. Economic scale is the significant factor affecting green high-quality development [28, 29]. Trade liberalization has a bigger im-pact on environmental quality in regions with larger economies. Thus, through the scale expansion effect, the creation of PFTZs can enhance the quality of local green economic growth.

The innovation-driven effect refers to the impact of PFTZs construction on the green high-quality development by influencing the regional innovation environment. According to Lee and Wang [29], lowering environmental pollution requires a conducive climate for innovation. With more palatable trade and investment rules, PFTZs draw in high-quality foreign money and expertise, enabling local businesses to innovate and expand while also indirectly enhancing the ecosystem for innovation. Lin and Ma [30] argued that the environment for innovation supports the flow of elements necessary for the development of green technologies, assisting in the balancing of eco-nomic quality and environmentally friendly development. As a result, PFTZs’ innovation-driven effect can enhance regional green high-quality development.

Based on the above analysis, this paper proposes the following research hypothesis.

Hypothesis 1 (H1): The establishment of PFTZs can significantly promote the regional green and high-quality development.

Hypothesis 2a (H2a): PFTZs can enhance green and high-quality development through the technological progress effect.

Hypothesis 2b (H2b): PFTZs can enhance green and high-quality development through the scale expansion effect.

Hypothesis 2c (H2c): PFTZs can enhance green and high-quality development through the innovation-driven effect.” which can be seen in Lines 136-184.

Point 3: The author lacks relevant secondary data, and the analysis results lack further analysis. Therefore, what kind of comparison between China and other countries can be analyzed and compared.

Response 3: Thank you for the comment and some corrections are made.

We examine the policy implications of the construction of PFTZs for China, with the aim of finding a suitable path for the sustainable development of China’s PFTZs. In the future study, we will further collect data on PFTZs in other countries for horizontal analysis and comparison.

Point 4: The overall research is a subjective conclusion, especially in the influence of "The Belt and Road Initiative" in fig 1, and the connection between the research has not been found at all, so there is a big gap between the actual research and the research structure.

Response 4: Thank you for the comment and some corrections are made.

According to your advice, we will look for solid theoretical support to explore the linkage between the PFTZs and the Belt and Road Initiative in the following stage of our research. We are currently enhancing and sublimating the original research theme, focusing on the key positioning of PFTZs in China’s new development pattern and analyzing the impact of PFTZs construction on China’s green and high-quality development.

Besides the above revisions, we do the following revisions.

First, we modify the indicator system by adding the level of economic development and environmental factors. Therefore, we sublimate green dual-circulation to green high-quality development based on the revision of the index system and the connotation of green dual circulation. Provided that data are available, we add two dimensions of economic development and environmental factors with six sub-indicators. The specific modifications are as follows.

“Green high-quality development is the explained variable. With the faster implementation of the new growth pattern, China’s economic and social development has given more attention to the quality and efficiency of green factor flows than most previous research on green high-quality development achieved [32, 34-36]. Therefore, this paper draws on the study of Wang et al. [37]to construct a green and high-quality development index (GHD) to represent the quality of regional green development by using the entropy weighting method. Table 1 shows the description and weights of relevant indicators.” which can be seen in Lines 223-229.

Second, we revise the title and abstract. The specific modifications are as follows.

Title: “Do China’s PFTZ promote green and high-quality development from the dual-circulation perspective”

Abstract: “Under the dual pressure of economic growth and environmental pressure, green and high-quality development become a critical strategy for China to achieve win-win development for both the economy and the environment. In the dual-circulation pattern, a deep and high level of openness is the primary prerequisite for encouraging green and high-quality development. Previous studies have overlooked the crucial pivotal role of PFTZs in the domestic and international circulation. The PFTZ is essential in driving China’s green modernization and high-quality development as a crucial area of opening up. This paper utilizes the propensity score matching and difference in difference (PSM-DID) on panel data from 27provinces from 2007 to 2020 to investigate the green policy effects of PFTZs building and its underlying mechanism. The results show that (1) the construction of PFTZs significantly enhance regional green and high-quality development, with a net policy impact of roughly 4%. The several robustness tests demonstrate that the outcomes are extremely reliable. (2) According to the heterogeneity analysis, compared with inland areas, coastal locations may more effectively foster green and high-quality development. (3) The influencing mechanism test results show that PFTZs primarily have positive impact on technological progress, scale expansion, and innovation-driven effects, which in turn support green and high-quality development. The mediating effect of innovation-driven and technological progress is extremely significant among them. These research findings can provide empirical support and policy rationale for PFTZs to better promote China’s green and high-quality development.”

Third, to make the introduction easier for the reader to understand, we revise it simultaneously. Since we add the theoretical analysis of the influencing mechanisms, we organize the literature review on PFTZs into the introduction section. 

Fourth, we revise the research methods section. Meanwhile, the original model has the issue with selective bias. Therefore, we employ the PSM-DID method for policy evaluation to get over this issue. Additionally, we modify the original model formulation. 

Fifth, we re-perform the empirical analysis due to the modification of the indicator system and research methodology. As a result, the empirical analysis section is significantly revised. For details, please see Section 4.

Sixth, we revise the conclusion section. The following are the specific revisions.

“As the link between domestic and foreign commerce, PFTZs acts as a driver for internal and external circulation and a crucial foundation for the advancement of high-quality green trade. Therefore, the construction of PFTZs is crucial to transform the economic development model and promote green and high-quality development. This study uses PSM-DID to evaluate the policy effects and underlying processes of PFTZs building on green and high-quality development based on panel data of Chinese provinces from 2007 to 2020. The results show that (1) the construction of PFTZs significantly enhance regional green and high-quality development, with a net policy impact of roughly 4%. The several robustness tests demonstrate that the outcomes are extremely reliable. (2) According to the heterogeneity analysis, compared with inland areas, coastal locations may more effectively foster green and high-quality development. (3) The influencing mechanism test results show that PFTZs primarily have positive impact on technological progress, scale expansion, and innovation-driven effects, which in turn support green and high-quality development. The mediating effect of innovation-driven and technological progress is extremely significant among them.

Based on the above conclusions, several policy insights for the sustainable growth of PFTZs might be provided.

First and foremost, the government and management of the relevant PFTZs should keep up their support for the development of new free trade zones, maintain a deep and high level of openness, and facilitate the flow of high-quality factors and re-sources. We should pay more attention to improving the level and quality of PFTZs and thus driving the green and high-quality development of the local economy, while we gradually expand the number of provincial free trade zones. we should focus more on promoting the local economy’s development of high standards and sustainability. Second, by determining the PFTZs’ functional placement, local businesses can be encouraged to avoid low-end industrialization, gradually eliminate environmentally polluting industries, and support the transition of the industrial structure to one that is more environmentally friendly. Inland PFTZs should also lessen the tax burden on businesses. Finally, according to the research in this paper, technological progress and green finance do not play a sufficient role in promoting green and high-quality development at this stage. PFTZs should therefore take note of technical innovation and breakthroughs. Furthermore, to improve the sustainability of regional economic development, PFTZs should create a comprehensive green finance system with green-oriented features.

This study recognizes the crucial function of PFTZs in the new pattern of development and establishes that PFTZs construction can effectively promote regional green and high-quality development. In other words, PFTZs can boost environmental advantages while enhancing the quality and effectiveness of factor flows. The research findings provide targeted policy recommendations for China’s PFTZs to support green and high-quality development. It also serves as a foundation for decision-making for the global development of PFTZs. There are still certain limitations, nevertheless, that need further development. We will analyze PFTZs and the countries along the Belt and Road in the future and thoroughly explore how China's PFTZs affect the green and high-quality development of other nations in light of the parallels between the two policies. Additionally, the spatial spillover effect on PFTZ building is a question worthy of more research.” which can be seen in Lines 352-361.

Lastly, more relevant papers and books are cited in the manuscript and are included in the References, which are listed below.

  1. BP statistical review of world energy 2022. London: British Petroleum Public Limited Company; 2022.
  2. Zhou G, Zhang Z, Fei Y. How to Evaluate the Green and High-Quality Development Path? An FsQCA Approach on the China Pilot Free Trade Zone. International Journal of Environmental Research and Public Health. 2022;19(1).
  3. Ye P, Zhang H, Ma S, Yang F, Li Y. A Knowledge Map Study of an Application of a Smart Land Planning Free-Trade Zone and China’s Contribution. Land. 2022;11(6).
  4. Zhao T, He F. Does the Pilot Free Trade Zone Promote the Quality of Urban Economic Growth: An Empirical Research Based on Quasi-Natural Experiment. Sustainability. 2022;14(12).
  5. Alvarez-Villa D, Guardado J. The long-run influence of institutions governing trade: Evidence from smuggling ports in co-lonial Mexico. Journal of Development Economics. 2020;144.
  6. Yang S, Martinez-Zarzoso I. A panel data analysis of trade creation and trade diversion effects: The case of ASEAN–China Free Trade Area. China Economic Review. 2014;29:138-51.
  7. Yanase A, Tsubuku M. Trade costs and free trade agreements: Implications for tariff complementarity and welfare. Interna-tional Review of Economics & Finance. 2022;78:23-37.
  8. Tachie AK, Xingle L, Dauda L, Mensah CN, Appiah-Twum F, Adjei Mensah I. The influence of trade openness on environmental pollution in EU-18 countries. Environmental Science and Pollution Research International. 2020;27(28):35535-55.
  9. He J. Pollution haven hypothesis and environmental impacts of foreign direct investment: The case of industrial emission of sulfur dioxide (SO) in Chinese provinces2. Ecological Economics. 2006;60(1):228-45.
  10. Wang L, Wang Z, Ma Y. Environmental Effects of Trade Liberalization–Evidence from China’s Pilot Free Trade Zone. Journal of International Commerce, Economics and Policy. 2021;12(03).
  11. Wang H, Zhang Y, Liu Z, Liu R, Li K. The impact and mechanisms of the Shanghai pilot free-trade zone on the green total factor productivity of the Yangtze River Delta Urban Agglomeration. Environmental Science and Pollution Research International. 2022;29(27):40997-1011.
  12. Erdogan AM. Bilateral trade and the environment: a general equilibrium model based on new trade theory. International Review of Economics & Finance. 2014;34:52-71.
  13. Li X, Huang Y, Li X, Liu X, Li J, He J, et al. How does the Belt and Road policy affect the level of green development? A quasi-natural experimental study considering the CO2 emission intensity of construction enterprises. Humanities and Social Sciences Communications. 2022;9(1):1-11.
  14. Zhang E, He X, Xiao P. Does Smart City Construction Decrease Urban Carbon Emission Intensity? Evidence from a Difference-in-Difference Estimation in China. Sustainability. 2022;14(23).
  15. Wang S, Gao S, Huang Y, Shi C. Spatiotemporal evolution of urban carbon emission performance in China and prediction of future trends. Journal of Geographical Sciences. 2020;30:757–74.
  16. Grossman GM, Krueger AB. Economic Growth and the Environment. The Quarterly Journal of Economics. 1995;110(2):353-77.
  17. Fei R, Cui A, Qin K. Can technology R&D continuously improve green development level in the open economy? Empirical evidence from China’s industrial sector. Environmental Science and Pollution Research. 2020;27:34052–66.
  18. Anser MK, Khan MA, Nassani AA, Aldakhil AM, Voo XH, Zaman K. Relationship of environment with technological in-novation, carbon pricing, renewable energy, and global food production. Economics of Innovation and New Technology. 2021;30(8):807-42.
  19. An T, Xu C, Liao X. The impact of FDI on environmental pollution in China: evidence from spatial panel data. Environmental Science and Pollution Research International. 2021;28(32):44085-97.
  20. Lee C-C, Wang F. How does digital inclusive finance affect carbon intensity? Economic Analysis and Policy. 2022;75:174-90.
  21. Lin B, Ma R. Green technology innovations, urban innovation environment and CO emission reduction in China: Fresh ev-idence from a partially linear functional-coefficient panel model2. Technological Forecasting and Social Change. 2022;176.
  22. Rosenbaum PR, Rubin DB. The Central Role of the Propensity Score in Observational Studies For Causal Effects. Biometrika. 1983;70(1):41-55.
  23. Xin B, Qu Y. Effects of Smart City Policies on Green Total Factor Productivity: Evidence from a Quasi-Natural Experiment in China. International Journal  of Environmental Research and Public Health. 2019;16(13).
  24. Beck T, Levine R, Levkov A. Big Bad Banks? The Winners and Losers from Bank Deregulation in the United States. Journal of Finance. 2010;65(5):1637-67.
  25. Wu H, Hao Y, Ren S. How do environmental regulation and environmental decentralization affect green total factor energy efficiency: Evidence from China. Energy Economics. 2020;91(C).
  26. Abbasab J, Sağsana M. Impact of knowledge management practices on green innovation and corporate sustainable development: A structural analysis. Journal of Cleaner Production. 2019;229:611-20.
  27. Fraccascia L, Giannoccaro I, Albino V. Green product development: What does the country product space imply? Journal of Cleaner Production. 2018;170:1076-88.
  28. Wang F, Wang R, He Z. Exploring the Impact of “Double Cycle” and Industrial Upgrading on Sustainable High-Quality Economic Development: Application of Spatial and Mediation Models. Sustainability. 2022;14(4).
  29. Baron RM, Kenny DA. The moderator-mediator variable distinction in social psychological research: conceptual, strategic, and statistical considerations. Journal of Personality and Social Psychology. 1986;51:1173-82.
  30. Acemoglu D, Akcigit U, Kerr W. Innovation network. Proceedings of the National Academy of Sciences of the United States of America. 2016; 113(41):11483-8.
  31. Anser MK, Khan MA, Awan U, Batool R, Zaman K, Imran M, et al. The Role of Technological Innovation in a Dynamic Model of the Environmental Supply Chain Curve: Evidence from a Panel of 102 Countries. Processes. 2020;8(9).
  32. He L, Liu R, Zhong Z, Wang D, Xia Y. Can green financial development promote renewable energy investment efficiency? A consideration of bank credit. Renewable Energy. 2019;143:974-84.
  33. Cui W, Li L, Chen G. Market-value oriented or technology-value oriented? ——Location impacts of indus-try-university-research (IUR) cooperation bases on innovation performance. Technology in Society. 2022;70.

Round 2

Reviewer 1 Report

1.The keywords should be reconsidered. In particular, the keyword "China" may be a little vague.

2.Please make sure the map is complete and acquired from an authoritative source.

Author Response

Point 1: The keywords should be reconsidered. In particular, the keyword "China" may be a little vague.

Response 1: Thank you for the comment and some corrections are made.

According to your advice, we revise the keywords.

“Keywords: pilot free trade zones; green and high-quality development; PSM-DID; innovation driven; quasi-natural experiment” which can be seen in Lines 26-27.

Point 2: Please make sure the map is complete and acquired from an authoritative source.

Response 2: Thank you for the comment and some corrections are made.

According to your advice, we add the note to explain the drawing information of Figure 1 in detail to ensure the integrity and authority of the map.

“Note: The authors map the distribution of each PFTZs according to the general scheme of PFTZ construction issued by the State Council. The map data are obtained from the National Platform for Common Geospatial Information Services.” which can be seen in Lines 64-66.

Besides the above revisions, we do the following revisions.

First, we revise the title to be more relevant to the study.

Title: “Does the establishment of PFTZs facilitate green and high-quality development from the dual-circulation perspective? Evidence from a quasi-natural experiment in China”

Second, we revise the theoretical analysis and supplement the relevant authoritative literature to provide detailed theoretical support. The specific modifications are as follows.

“The Environment Kuznets Curve (EKC) hypothesis is the classic theoretical model for measuring the relationship between economic development and environmental quality. Researchers summarize the intrinsic influence mechanism of the EKC hypothesis based on its viability [25-27]. Specifically, scale, structure, and technology effects of a region’s economic development have an impact on that region’s environmental quality. While, under free trade, the dynamic link between economic activity and environmental quality is even more pronounced [28]. The main goal of establishing PFTZs is to provide China with the expertise necessary to create a new open economic system through system innovation. Therefore, promoting trade liberalization is the key task for the construction and development of PFTZs. Based on this, EKC hypothesis put forward by Grossman and Krueger [29] can be used to explain the theoretical mechanism of PFTZs building on regional green and high-quality development. According to the functional positioning of PFTZs in the dual-circulation development pattern, this paper integrates EKC theory and related scholars’ researches to analyze the internal mechanism of policy effects from three paths: technological progress, scale expansion and innovation-driven effects.” which can be seen in Lines 143-157.

Third, more relevant papers and books are cited in the manuscript and are included in the References, which are listed below.

  1. Ma X, Jiang Q. How to Balance the Trade-off between Economic Development and Climate Change? Sustainability. 2019;11(6).
  2. Qian Z, Zhao Y, Shi Q, Zheng L, Wang S, Zhu J. Global value chains participation and CO emissions in RCEP countries2. Journal of Cleaner Production. 2021;332.
  3. Zhao C, Wang K, Dong X, Dong K. Is smart transportation associated with reduced carbon emissions? The case of China. Energy Economics. 2022;105.
  4. Halicioglu F. An econometric study of CO emissions, energy consumption, income and foreign trade in Turkey2. Energy Policy. 2009;37(3):1156-64.

Fourth, some small revisions are also made to make the expressions in the manuscript more precise and fluent, which could be seen in Line 12, Line47, Lines 76-77, Lines 226-227.

Reviewer 2 Report

The author carefully revised their manuscript. Fortunately, the quality of this manuscript has improved significantly. Congratulations!

Author Response

Thank you for your positive comments and valuable suggestions to improve the quality of our manuscript.

Compared to the original manuscript, we revise the following parts.

First, we revise the title and keywords to be more relevant to the study.

Title: “Does the establishment of PFTZs facilitate green and high-quality development from the dual-circulation perspective? Evidence from a quasi-natural experiment in China”

“Keywords: pilot free trade zones; green and high-quality development; PSM-DID; innovation driven; quasi-natural experiment” which can be seen in Lines 26-27.

Second, we add the note to explain the drawing information of Figure 1 in detail to ensure the integrity and authority of the map.

“Note: The authors map the distribution of each PFTZs according to the general scheme of PFTZ construction issued by the State Council. The map data are obtained from the National Platform for Common Geospatial Information Services.” which can be seen in Lines 64-66.

Third, we revise the theoretical analysis and supplement the relevant authoritative literature to provide detailed theoretical support. The specific modifications are as follows.

“The Environment Kuznets Curve (EKC) hypothesis is the classic theoretical model for measuring the relationship between economic development and environmental quality. Researchers summarize the intrinsic influence mechanism of the EKC hypothesis based on its viability [25-27]. Specifically, scale, structure, and technology effects of a region’s economic development have an impact on that region’s environmental quality. While, under free trade, the dynamic link between economic activity and environmental quality is even more pronounced [28]. The main goal of establishing PFTZs is to provide China with the expertise necessary to create a new open economic system through system innovation. Therefore, promoting trade liberalization is the key task for the construction and development of PFTZs. Based on this, EKC hypothesis put forward by Grossman and Krueger [29] can be used to explain the theoretical mechanism of PFTZs building on regional green and high-quality development. According to the functional positioning of PFTZs in the dual-circulation development pattern, this paper integrates EKC theory and related scholars’ researches to analyze the internal mechanism of policy effects from three paths: technological progress, scale expansion and innovation-driven effects.” which can be seen in Lines 143-157.

Fourth, more relevant papers and books are cited in the manuscript and are included in the References, which are listed below.

  1. Ma X, Jiang Q. How to Balance the Trade-off between Economic Development and Climate Change? Sustainability. 2019;11(6).
  2. Qian Z, Zhao Y, Shi Q, Zheng L, Wang S, Zhu J. Global value chains participation and CO emissions in RCEP countries2. Journal of Cleaner Production. 2021;332.
  3. Zhao C, Wang K, Dong X, Dong K. Is smart transportation associated with reduced carbon emissions? The case of China. Energy Economics. 2022;105.
  4. Halicioglu F. An econometric study of CO emissions, energy consumption, income and foreign trade in Turkey2. Energy Policy. 2009;37(3):1156-64.

Fifth, some small revisions are also made to make the expressions in the manuscript more precise and fluent, which could be seen in Line 12, Line47, Lines 76-77, Lines 226-227.

Reviewer 3 Report

#1 "Do Pilot Free Trade Zones Promote Green Dual-Circulation Development? -a Quasi-Natural Experimental Evidence from China", but such a title seems to be far from what the research content wants to express.

#2 The author still lacks academic support for relevant hypotheses, so the overall analysis process cannot present objective source evidence.

#3 The author still avoids the blind spots of the research, and if it cannot be dealt with accurately, the research scope and topics must be narrowed.

#4 Well done than the previous version. But the author must think about the overall revision content of the first three questions so as to meet and match what you want to study.

Author Response

Point 1: "Do Pilot Free Trade Zones Promote Green Dual-Circulation Development? -a Quasi-Natural Experimental Evidence from China", but such a title seems to be far from what the research content wants to express.

Response 1: Thank you for the comment and some corrections are made.

We confirm the title in the revised manuscript. After communicating with the editor, we understand that the editor will make systematic changes based on the manuscript. Therefore, we ensure that the title of the manuscript is revised correctly.

Title: “Does the establishment of PFTZs facilitate green and high-quality development from the dual-circulation perspective? Evidence from a quasi-natural experiment in China”

Point 2: The author still lacks academic support for relevant hypotheses, so the overall analysis process cannot present objective source evidence.

Response 2: Thank you for the comment and some corrections are made.

The theoretical basis of the research hypothesis in this paper is the EKC theory proposed by Grossman and Krueger. This theory is mainly used to explain the relationship between economic activities and environmental pollution in the free trade. Promoting trade liberalization is one of the main objectives of the PFTZs’ building and development. Therefore, there is certain academic support for the research hypothesis of this paper based on EKC theory. Meanwhile, according to the characteristics of PFTZs in dual-circulation development, this paper extends the EKC theory to analyze the impact of PFTZs on green and high-quality development in terms of three mechanisms, namely, technological progress, scale expansion and innovation drive.

According to your advice, we revise the theoretical analysis and supplement the relevant authoritative literature to provide detailed theoretical support. The specific modifications are as follows.

“The Environment Kuznets Curve (EKC) hypothesis is the classic theoretical model for measuring the relationship between economic development and environmental quality. Researchers summarize the intrinsic influence mechanism of the EKC hypothesis based on its viability [25-27]. Specifically, scale, structure, and technology effects of a region’s economic development have an impact on that region’s environmental quality. While, under free trade, the dynamic link between economic activity and environmental quality is even more pronounced [28]. The main goal of establishing PFTZs is to provide China with the expertise necessary to create a new open economic system through system innovation. Therefore, promoting trade liberalization is the key task for the construction and development of PFTZs. Based on this, EKC hypothesis put forward by Grossman and Krueger [29] can be used to explain the theoretical mechanism of PFTZs building on regional green and high-quality development. According to the functional positioning of PFTZs in the dual-circulation development pattern, this paper integrates EKC theory and related scholars’ researches to analyze the internal mechanism of policy effects from three paths: technological progress, scale expansion and innovation-driven effects.” which can be seen in Lines 143-157.

Point 3: The author still avoids the blind spots of the research, and if it cannot be dealt with accurately, the research scope and topics must be narrowed.

Response 3: Thank you for the comment and some corrections are made.

According to the functional positioning of China’s PFTZs in the dual-circulation development, this paper utilizes the PSM-DID method to test the policy effects and intrinsic mechanisms of PFTZs construction on China’s green and high-quality development. Based on data availability, this paper focuses mainly on assessing the policy impact of China’s PFTZs construction. Meanwhile, according to the characteristics of regional heterogeneity, this paper analyzes the policy differences of building PFTZs in different regions. Additionally, on the support of EKC theory, this paper puts forward the research hypothesis that PFTZs promote green and high-quality development through three paths: technological progress, scale expansion and innovation-driven effects, and verifies the research hypothesis through the mediating effect model.

Point 4: Well done than the previous version. But the author must think about the overall revision content of the first three questions so as to meet and match what you want to study.

Response 4: Thank you for the comment and some corrections are made.

Based on the literature review and theoretical analysis, this paper poses three research questions. Will the creation of PFTZs considerably enhance green and high-quality development? Does the impact of PFTZs construction on green policies vary by region? What is the influencing mechanism by which PFTZs building affects green and high-quality development? Based on the above three questions, this paper conducts an empirical analysis using PSM-DID and mediating effect models, respectively. Following are the conclusions drawn in this paper based on the validation. (1) The establishment of PFTZs significantly enhance regional green and high-quality development, with the net policy impact of roughly 4%. The several robustness tests demonstrate that the outcomes are extremely reliable. (2) According to the heterogeneity analysis, compared with inland areas, coastal locations may more effectively foster green and high-quality development. (3) The influencing mechanism test results show that PFTZs primarily have positive impact on technological progress, scale expansion, and innovation-driven effects, which in turn support green and high-quality development. The mediating effect of innovation-driven and technological progress is extremely significant among them. These findings correspond to the questions and satisfy the research that we want to explore in this paper.

Besides the above revisions, we do the following revisions.

First, we revise the keywords to be more relevant to the study.

“Keywords: pilot free trade zones; green and high-quality development; PSM-DID; innovation driven; quasi-natural experiment” which can be seen in Lines 26-27.

Second, we add the note to explain the drawing information of Figure 1 in detail to ensure the integrity and authority of the map.

“Note: The authors map the distribution of each PFTZs according to the general scheme of PFTZ construction issued by the State Council. The map data are obtained from the National Platform for Common Geospatial Information Services.”which can be seen in Lines 64-66.

Third, more relevant papers and books are cited in the manuscript and are included in the References, which are listed below.

  1. Ma X, Jiang Q. How to Balance the Trade-off between Economic Development and Climate Change? Sustainability. 2019;11(6).
  2. Qian Z, Zhao Y, Shi Q, Zheng L, Wang S, Zhu J. Global value chains participation and CO emissions in RCEP countries2. Journal of Cleaner Production. 2021;332.
  3. Zhao C, Wang K, Dong X, Dong K. Is smart transportation associated with reduced carbon emissions? The case of China. Energy Economics. 2022;105.
  4. Halicioglu F. An econometric study of CO emissions, energy consumption, income and foreign trade in Turkey2. Energy Policy. 2009;37(3):1156-64.

Fourth, some small revisions are also made to make the expressions in the manuscript more precise and fluent, which could be seen in Line 12, Line47, Lines 76-77, Lines 226-227.

Round 3

Reviewer 3 Report

#1 The title of "Does the establishment of PFTZs facilitate green and high-quality development from the dual-circulation perspective? Evidence from a quasi-natural experiment in China Yanjun Chang" seems to be far from the scope of the research topic.

Well done than the previous version in other inquires from the reviewer.

Author Response

Point 1: The title of "Does the establishment of PFTZs facilitate green and high-quality development from the dual-circulation perspective? Evidence from a quasi-natural experiment in China Yanjun Chang" seems to be far from the scope of the research topic.

Response 1: Thank you for the comment and some corrections are made.

According to your advice, we revise the title to match the content of the study. The purpose of this paper is to evaluate the policy effects of China’s PFTZs construction on green and high-quality development and its underlying mechanisms based on the dual-circulation perspective. In light of this, we revise the title. The details are as follows.

Title: “Effects and mechanisms of China’s pilot free trade zones on green and high-quality development from the dual-circulation perspective”
